# Atlantic water recirculation in the northern Barents Sea affects winter sea ice extent

Finn Ole Heukamp [1] ✉, Claudia Wekerle [1], Torsten Kanzow [1,2], Rebecca McPherson [1] & Till M. Baumann[3,4]

Over the past 50 years, Arctic sea ice has declined in all seasons, with particularly pronounced winter reductions in the Barents Sea. While temperature changes in the Atlantic Water inflow and atmospheric-driven melt have been identified as key drivers of this decline, the role of the return-flow of Atlantic Water in the northern Barents Sea Opening, linked to its recirculation back into the Nordic Seas, has remained largely unrecognized. Using a global ocean and sea ice model, we find that the volume transport of the Atlantic Water return-flow is strongly correlated with the sea ice area in the Barents Sea. In addition, we find that, over the past 40 years, the return-flow has steadily weakened without a corresponding change in inflow. Here, we show that reduced Atlantic Water removal by a weakened return-flow contributes to both interannual variability and the sustained loss of Barents Sea sea ice.

In the Barents Sea (BS), one of two Atlantic Ocean gateways to the Arctic Ocean, observations have revealed a substantial retreat of winter sea ice cover[1–4] superposed on pronounced interannual variability[4] that may periodically reverse the overall sea ice decline[5]. The long-term decline is ascribed to the overall increased ocean heat transport[3], which is mainly due to the warming of the inflowing water. Furthermore, local feedbacks have been suggested to contribute to the accelerated loss of sea ice in the BS[6,7]. The interannual variability of the BS sea ice has, however, been ascribed to variations in

- ocean heat content, driven, with a lag of 1 year, by the varying ocean heat transport through the western Barents Sea Opening (BSO)[3,5,8–14],
- wind-driven sea ice drift through the eastern and northern gates connecting the BS to the Arctic Ocean and the Kara Sea[4,15–18], and
- atmospheric processes, such as pronounced Ural blocking[19], La Niña events[20], regional anticyclonic wind anomalies[21], and air temperature fluctuations[22].

However, the relative importance of these oceanic and atmospheric processes and the time scales on which they occur remain unclear[23]. Here, an additional oceanic contribution to sea ice variability in the BS is proposed.

The BS plays a key role in transporting ocean heat from the Atlantic Ocean to the Arctic Ocean[8,9,24] and is thus of particular interest. It is connected to the Nordic Seas via the BSO, where warm and saline Atlantic Water (AW), which is the dominant source of ocean heat of the BS, enters from the west[25,26] (Fig. 1a). The inflow of AW into the BS occurs in the southern and central parts of the BSO through the Norwegian Coastal Current (NCC) and the central AW inflow. In the northern part, cooled and freshened AW is exported back into the Nordic Seas by a return-flow[27], which is fed by a recirculation of AW in the BS[7,27–29] (Fig. 1a).

The AW inflow into the BSO has been monitored by the Norwegian Institute of Marine Research since 1997, which has maintained an array consisting of 5 moorings designed to capture the central AW inflow into the BS, carrying most of the AW heat[9] across 19. 7°E between 71. 5°N and 73. 5°N[3,25,26] (Fig. 1b). In terms of the volume transport, the NCC carries 1.2Sv into the BS[9] and the central AW inflow carries 2.3Sv into the BS[3]. The interannual variability (standard deviation (STD) of the annual means) of the central AW inflow is estimated at 0.4Sv[3].

The return-flow in the northern BSO is less well documented, as mooring deployments there are particularly risky due to intense fishery activity in this region. Nevertheless, the return-flow was monitored over 2 years when an additional mooring had been deployed at 19.25°E, 73.85°N from September 2003 to October 2005[28] (Fig. 1b). Based on

[1]Alfred-Wegener-Institut, Helmholtz Centre for Polar and Marine Research, Bremerhaven, Germany. [2]University of Bremen, Bremen, Germany. [3]Institute of Marine Research, Bergen, Norway. [4]Bjerknes Centre for Climate Research, Bergen, Norway. ✉e-mail: finn.heukamp@awi.de

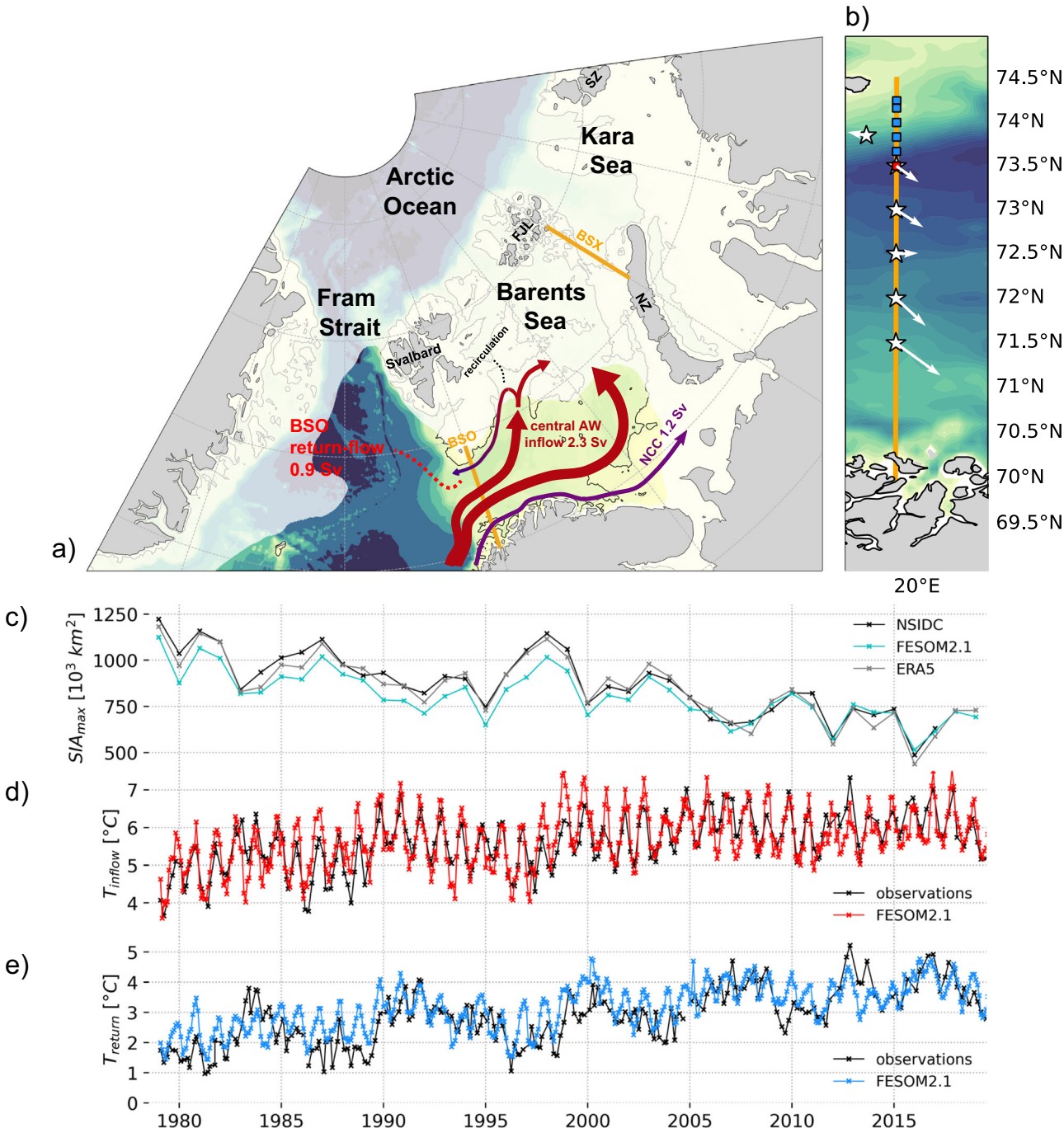

**Fig. 1 | Atlantic Water pathways towards the Arctic Ocean and model performance in the Barents Sea. a** Schematic of Atlantic Water (AW) pathways into the Barents Sea (BS) and towards the sea ice. The color shading depicts ocean depth from deep (blue) to shallow (yellow). White shading qualitatively indicates winter sea ice cover. Abbreviations: BSO Barents Sea Opening, BSX Barents Sea Exit, NZ Novaya Zemlya, FJL Franz-Joseph-Land, SZ Severnaya Zemlya, AW Atlantic Water, NCC Norwegian Coastal Current. The transports are observational estimates. **b** Map of mooring (stars) and Conductivity Temperature Depth (CTD) locations (squares) along the BSO used in this study. White quivers represent the average velocity at the mooring locations (50 m for the inflow moorings, 50–200 m for the return-flow mooring). **c** Annual maximum sea ice area in the BS (18–60°E, 68–81°N) based on NSIDC satellite data, ERA5 reanalysis, and model output from FESOM2.1. **d** AW temperature in the central AW inflow of the BSO at 73°N (50–200 m) derived from regular conductivity temperature depth (CTD) stations (red square in **b**) and FESOM2.1. **e** Water temperature in the return-flow of the BSO between 73.67°N and 74.25°N (50 m–bottom, see "Methods") was derived from regular CTD transects (blue squares in **b**) and FESOM2.1. Note that all temperature observations are snapshots taken roughly six times a year while the model provides monthly averages.

this mooring it was estimated that the return-flow carries 0.9Sv (annual mean) out of the BS toward Fram Strait[28] of which ~80% represents AW[27]. No estimates of the return-flow's interannual variability have, however, been acquired due to the lack of multi-year observations. Based on a numerical simulation, the winter volume transport of the

return-flow and its variability has been estimated as $1.6 \pm 0.25$Sv[29] (1970–2019), which appears high compared to the (limited) observations, but may in part be explained by the generally strong winter intensification of the velocity field in the BSO[25,26,28] and the pronounced downward trend over the entire simulated period[29]. These previous

findings, however, suggest that the return-flow may significantly contribute to the interannual variability of the net AW transport through the BSO, which is yet unaccounted for in observational BSO AW transport estimates, which only consider the AW inflow regions[9,13,24]. In regard to this variability, barotropic transport anomalies of the return-flow are mainly driven by air pressure anomalies over Svalbard[7,28,29]. Associated (anti-)cyclonic wind anomalies modify the divergence of the Ekman transport onto or off the northern BS/Svalbard shelf. In turn, the meridional sea surface height gradient in the northern BSO, which drives the return-flow, is modified[7]. In addition, the warm and saline AW south of the return-flow and the cold and fresh Polar Water north of the return-flow result in a strong meridional density gradient across the return-flow, adding a pronounced baroclinic component to the velocity field, which is reflected in a bottom-intensification of the flow.

With its substantial transport and variability, the return-flow may have significant impacts on the regional climate system. These stretch from affecting sea ice in the BS via modified AW transport, to altering the water mass properties of the West Spitsbergen Current in Fram Strait, the second major ocean gateway to the central Arctic Ocean. Associated co-variability of the BS return-flow and the Fram Strait AW transport has already been identified[28].

In this study, we show that the westward return-flow in the northern BSO strongly contributes to the interannual net AW transport variability through the BSO, based on a well-evaluated high-resolution numerical hindcast simulation with the Finite volumE Sea Ice Ocean Model (FESOM2.1) and the GLORYS12V1 (GLORYS) ocean reanalysis. In a second step, we examine the particular impact of variations of the AW return-flow on the interannual variability of sea ice and its negative trend in the BS.

## Results

### Westward return-flow in the northern BSO as a key driver of Atlantic Water transport variability

For this study, we conducted a hindcast simulation with a configuration of FESOM2.1 that has been optimized for the Nordic Seas and Arctic Ocean. The simulation covers the period 1958–2019, of which the period 1979–2019 is examined in this study (see "Methods" for model details). The model demonstrates high skill in reproducing observed maximum annual sea ice area ($SIA_{max}$) variability in the BS (correlation: $R = 0.92$, $p \leq 0.01$) (Fig. 1c) and AW temperature variability (Fig. 1d, e) in the BSO AW inflow (monthly: $R = 0.78$, winter: $R = 0.59$, $p \leq 0.01$) and return-flow (monthly: $R = 0.57$, winter: $R = 0.74$, $p \leq 0.01$). It should, however, be noted that the temperature observations depict irregularly conducted snapshots in time, whereas the model provides monthly means. Given the model's skill in reproducing the aforementioned properties, we further examine the variability of the AW volume transports using the model.

In the BSO, between Norway and Bear Island, only the return-flow in the northern BSO moves AW westward (Fig. 1a, b). To better understand the role of the return-flow in the net AW volume transport and its interannual variability, we divided the simulated volume transports into components contributing to the eastward flow (NCC and central AW inflow) and those contributing to westward flow (return-flow) by separating the flow by its direction (eastwards or westwards). For the 1979–2019 period, the results generally show that the volume transport of AW inflow is stronger than that of the return-flow, with annual means of 0.9Sv (NCC), 2.8Sv (central AW inflow), and 1.2Sv (return-flow)—a result consistent with observational estimates[3,27].

As we later turn our focus on examining winter conditions, particularly the annual sea ice maximum in early spring, in the following, all numbers presented refer to winter means, i.e., December to March averages. Notably, volume transports through the BSO typically peak during winter[26,27], and so does the total AW inflow in the model with a winter mean transport of 4.5Sv (Fig. 2a). The return-flow (Fig. 2b), however, is slightly weaker in winter (1.0Sv) than on annual average in the model simulation.

The interannual variability of the return-flow volume transport, however, presents a picture distinct from the dominance of the AW inflow in the total transport of AW through BSO. The return-flow exhibits remarkably high interannual variability equivalent to ~74% of the variability in AW inflow (standard deviations of the linearly detrended data: inflow: 0.42Sv, return-flow: 0.31Sv). In some cases, return-flow volume transport anomalies even surpass those of the inflow (Fig. 2c), demonstrating the return-flow's ability to significantly add variability to the winter AW transports through the BSO.

In order to put our model results on a broader basis, especially with regard to the simulated volume transports of the return-flow, which cannot be evaluated with in-situ measurement data, we also make use of the ocean reanalysis GLORYS12V1 (GLORYS), available from 1993 onward. With a horizontal resolution of about 8 km, GLORYS has only half the resolution of FESOM2.1, but the assimilation of in-situ data and especially satellite altimetry data allows an estimation of the performance of the model. For the 1993–2019 period, in which model and reanalysis are both available, the winter mean volume transport of the return-flow in the ocean reanalysis is slightly weaker (model: 0.85Sv, reanalysis: 0.56Sv) (Fig. 2b). In terms of interannual variability of the return-flow's volume transport, both model and reanalysis indeed reveal similar orders of magnitude (standard deviations of the linearly detrended data 1993–2019: model: 0.36Sv, reanalysis: 0.32Sv) and are further strongly correlated ($R = 0.72$, $p \leq 0.01$) (Fig 2b, d). The consistency between the model and reanalysis provides confidence in the model's ability to simulate the interannual transport variability of the return-flow in the BSO.

In terms of the AW, it is not only the volume transport but also the temperature of the AW that is of particular interest. Temperature differences further differentiate the two flows: in the model, the inflow is warmer ($5.4 \pm 0.24\,°C$) and more stable compared to the cooler ($3.0 \pm 0.50\,°C$) but much more variable return-flow (Fig. 1d, e; Fig. SI1). While the winter mean inflow and return-flow temperatures are strongly correlated ($R = 0.80$, $p \leq 0.01$), suggesting rapid recirculation on seasonal timescales, their respective volume transport variabilities are essentially uncorrelated ($R = 0.23$, $p \leq 0.01$), (Fig. 2c). This lack of correlation suggests distinct underlying mechanisms governing the AW inflow and return-flow. With AW representing the dominant water mass in the return-flow (80%)[27], and its magnitude of interannual variability that is of similar order or even exceeds that of the inflow in both volume and temperature, the model simulation suggests the return-flow plays a pronounced role in determining the net AW volume transport variability in the BSO, which is as yet not accounted for. Since the variability of the AW transport through the BSO plays a role in the extent of sea ice, the strong influence of the return-flow on the interannual transport variability of AW shown by the model inevitably raises the question of its influence on sea ice variability in the BS.

### A link between the Atlantic Water return-flow in the Barents Sea Opening and sea ice area in the Barents Sea

To assess the ability of the return-flow in the BSO to contribute to $SIA_{max}$ variability, we compare the time series of $SIA_{max}$ in the BS, usually reached in March or April, with the time series of the winter mean volume transport of the return-flow (Fig. 2c). Indeed, there is a robust correlation between anomalous return-flow volume transport and anomalous $SIA_{max}$ ($R = 0.71$, $p \leq 0.01$). Additionally, both the return-flow and $SIA_{max}$ reveal a robust downward trend during the 1979–2019 period. While the winter mean volume transport of the return-flow weakened by 0.019Sv per year (or 0.76Sv over the past 40 years, representing a sizable fraction of the time mean transport), $SIA_{max}$ retreated by ~8000 km$^2$ per year (Figs. 2b and 1c). Both the trends and covariability of $SIA_{max}$ and the return-flow point towards a contribution of the return-flow to the net AW transport

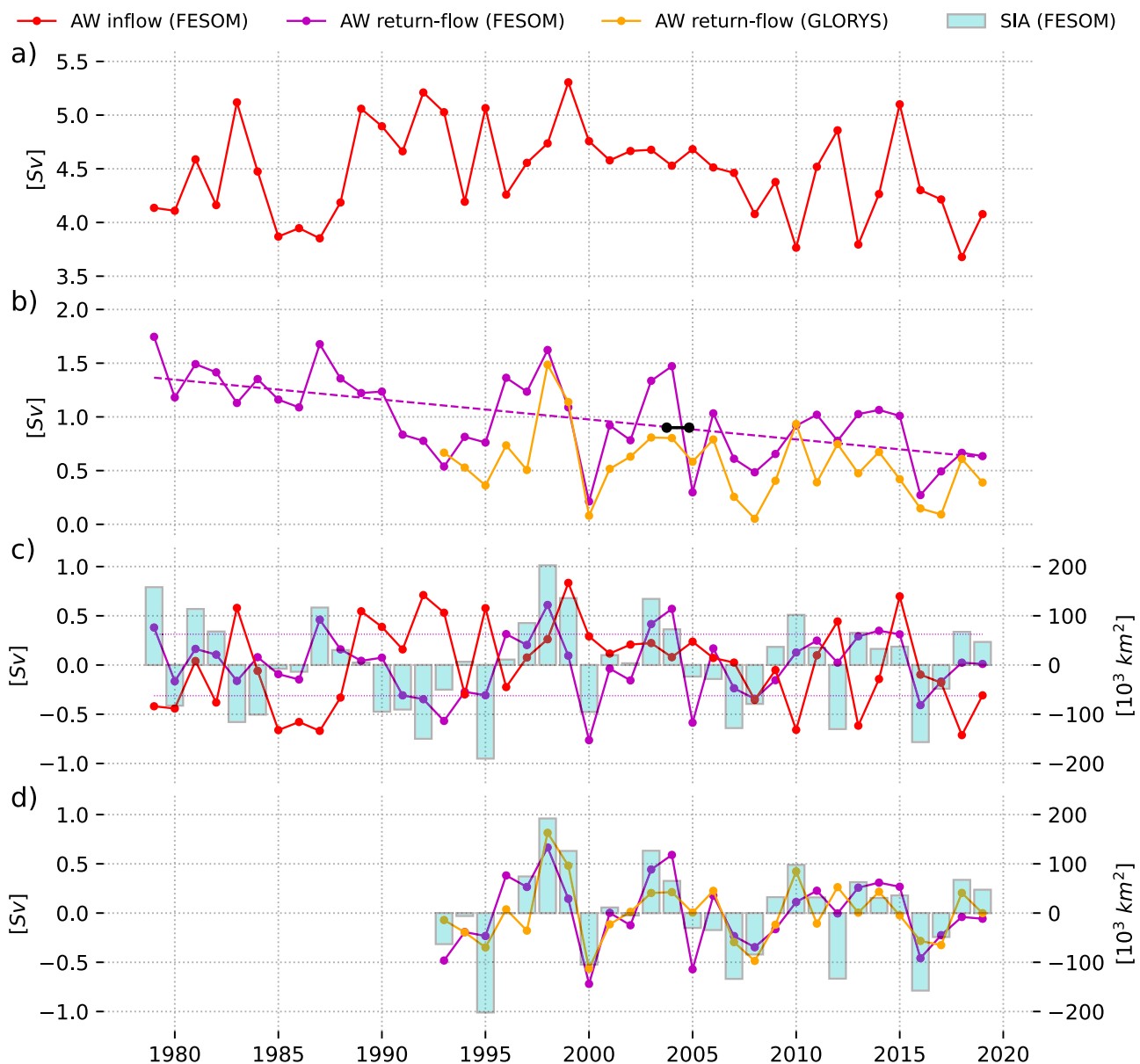

**Fig. 2 | Atlantic water flow and sea ice area variability in the Barents Sea.**
**a** Winter volume transport (December-March mean) of the Atlantic Water (AW)
inflow and (**b**) of the return-flow for period 1979–2019 (FESOM2.1) and 1993-2019
(GLORYS). Note that positive numbers in (**b**) indicate an increased return-flow.
Statistically significant linear trends (99% confidence) are shown in (**b**) as dashed
lines. Black line indicates annual mean volume transport estimated from the 2-year
mooring period. **c** Anomalous and detrended AW inflow, return-flow, and max-
imum annual Barents Sea sea ice area (cyan bars, right axis) for the period
1979–2019. Dotted magenta lines depict the standard deviation of the anomalous
return-flow volume transport. **d** Anomalous and detrended return-flow for
FESOM2.1 and GLORYS for the period 1993–2019.

variability through the BSO, which, in turn, affects the winter sea ice.
The temperature of the return-flow is, in contrast, weakly anti-
correlated with $SIA_{max} (R = -0.42, p \leq 0.01)$.

Interestingly, there is no significant zero-lag correlation between
the time series of winter mean AW inflow volume transport and
$SIA_{max} (R = 0.29, p \leq 0.01)$. Nor is there a statistically significant corre-
lation ($p \leq 0.01$) when correlating $SIA_{max}$ with the individual contribu-
tions to the AW inflow (namely the NCC and the central AW inflow),
even when accounting for a 1-year lag between the return-flow volume
transport and $SIA_{max}$. Furthermore, in contrast to the return-flow, no
trend is present in the volume transport of the AW inflow (Fig. 2a).

To support our correlation analysis and to obtain a spatial view of
changes in the BSO velocity field associated with anomalous $SIA_{max}$ in
the BS, particularly the return-flow, we apply a linear least-squares
regression analysis. Here, the time series of the winter mean zonal

velocity in each grid cell in the depth-latitude section of the BSO (Fig. 3a)
are taken as predictors for the time series of the annual $SIA_{max}$. The
highest regression coefficients are found in the return-flow over
the slope of Svalbard Bank in the northern BSO (Fig. 3b). Outside of the
return-flow, the regression is not significant. Our regression analysis
suggests an increased barotropic component of the return-flow's velo-
city field in winters being followed by increased $SIA_{max}$ in the BS. Given
the time-mean baroclinic, bottom-intensified structure of the return-
flow, the barotropic structure emerging from the regression fit points
towards a wind-driven rather than a density-driven modification of the
current. This is further supported by a positive sea level air pressure
anomaly over the north-western BS emerging in winters with pro-
nounced return-flow (Fig. SI2a). The associated anomalous anticyclonic
winds drive Ekman transport onto Svalbard Bank, ultimately controlling
the sea surface height gradient, which determines the barotropic

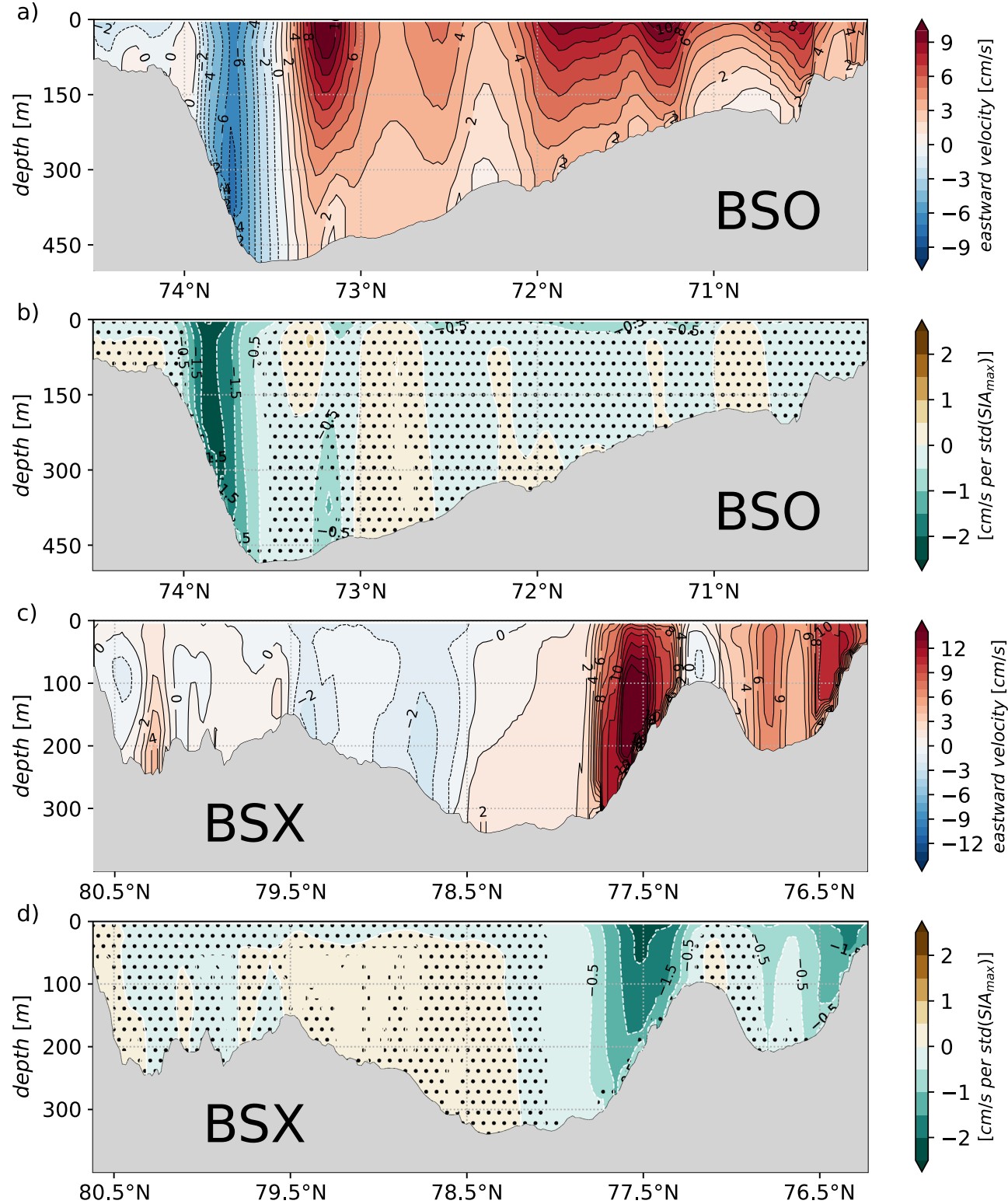

**Fig. 3 | Velocity transects of the Barents Sea Opening and Barents Sea Exit.**
**a** Winter mean eastward velocity (red/positive denotes inflow, blue/negative denotes outflow) along the Barents Sea Opening (BSO) in FESOM2.1 from 1979 to 2019. **b** Regression coefficients where the annual winter mean eastward velocity in the BSO is taken as a predictor for the annual maximum sea ice area of the Barents Sea of the same winter. Dotted areas indicate non-significant regression coefficients (99% confidence). **c**, **d** as in **a**, **b** for the Barents Sea Exit (BSX) along 60˚E (Fig. 1).

component of the AW return-flow. In this regard, the general weakening of the return-flow coincides with a slight trend in sea level air pressure (~−1hPa/decade) over the Svalbard shelf (Fig. SI2b, c).

The two branches of the AW inflow into the BS (Fig. 1a) are clearly represented in the simulation (Fig. 3a), with the central AW inflow residing between 71. 5°N and 73. 5°N and the NCC close to the Norwegian coast. Neither branch, however, depicts a linear relationship to the SIA$_{max}$ given the non-significant regressions (Fig. 3b). For reasons of mass conservation, any transport anomaly of the return-flow requires a near-instantaneous compensating change in the volume

transport somewhere else across the boundary of the BS. This compensation is not provided by the inflow through the BSO (Fig. 3b) as already noted by the lack of correlation. Consequently, the modified return-flow must be balanced downstream of the AW pathway through the BS, most likely in the Barents Sea Exit (BSX) section between Novaya Zemlya and Franz-Joseph-Land (Fig. 1a), where the strongly cooled, modified AW leaves the BS as subsurface water mass[12,14].

Indeed, the eastward volume transport through the BSX (from the BS into the Kara Sea) is strongly anti-correlated with $SIA_{max}$ ($R = -0.69, p \leq 0.01$). Repeating the regression analysis (Fig. 3b) with the velocity field along the BSX section reveals that the outflow out of the BS across the BSX section in winter (Fig. 3c) is weaker in winters with increased $SIA_{max}$ (Fig. 3d). In contrast to the almost barotropic anomaly of the return-flow in the BSO (Fig. 3b), the regression analysis in the BSX reveals a surface-intensified response (Fig. 3d). This component of the flow yields the mass-compensating mechanism of anomalous return-flow events (Fig. SI3). Our results thus demonstrate a barotropically strengthened westward return-flow in the BSO that is compensated by a baroclinic, upper ocean intensified, weakening of the eastward volume transport through the BSX. The quasi-simultaneous velocity/volume transport co-variability between the return-flow in the BSO and the currents in the BSX therefore suggests a connection along the AW pathways through the BS, which is further explored in the following section.

### Quasi-simultaneous flow variability along the Atlantic Water pathways affects sea ice in the Barents Sea

Since we are generally interested in how the AW circulation anomalies affect sea ice area in the wintertime, we proceed to visualize the spatial structure of the suggested compensatory mechanism between transport anomalies of the BSO return-flow and those through BSX provided by the AW circulation through the BS. For this, we apply a composite analysis and extract the winter mean upper ocean velocity anomalies from the model run, separately averaging data from the years in which the detrended return-flow volume transport either exceeds or is less than the time mean by one STD (Fig. SI4). We choose a depth range of 5–50 m for the analysis, which is shallow enough so that water from this level will supply heat to reduce sea ice formation in wintertime. At the same time, the level is deep enough so that it should represent the larger-scale geostrophic flow regime of AW in the BS. It further accounts for the near-surface intensification of the flow, as demonstrated in the BSX regression analysis (Fig. 3d). The results reveal coherent patterns of flow variability along the AW circulation pathways connecting the BSO and the BSX (Figs. 3b, d and 4a, b). Specifically, our simulation indicates that in winters with increased return-flow the velocities along both the northern and the more pronounced southern AW pathway through the BS east of roughly 35°E are decelerated, while the recirculation mainly happening west of 35°E and supplying the BSO return-flow with AW is strengthened (Fig. 4a). This shows that AW transport towards the central and western BS is reduced as AW is instead redirected out of the BS by the recirculation and the return-flow. As a consequence of these circulation changes, a pronounced negative anomaly of the upper ocean heat content is found in the central and western BS in winters with pronounced return-flow (Fig. 4c). The reduced central BS ocean heat content is further reflected in increased sea ice concentration (Fig. 4e). In years with decreased return-flow all anomalies are of opposite sign (Fig. 4b, d, f), meaning accelerated circulation through the BS (Fig. 4b), increased upper ocean heat content (Fig. 4d) and reduced sea ice concentration (Fig. 4f). Hence, our results indicate that pronounced AW recirculation and return-flow cause an upper ocean in the central BS that is anomalously cold, as a decent fraction of the AW heat is removed from the BS, facilitating formation and persistence of sea ice and thus allowing increased $SIA_{max}$. In contrast, in weak return-flow years, the AW heat penetrates deep into the BS, facilitating sea ice melt and thus reducing $SIA_{max}$.

Given the high spatial variability of sea ice in the BS[4], we attempt to identify the specific area in which the impact of the AW circulation on the SIA is strongest. Thus, we replace the single time series of $SIA_{max}$ accounting for the entire BS with a suite of $SIA_{max}$ time series obtained from spatially overlapping areas within the BS (methods). We then proceed by computing Pearson correlations between the time series of winter mean volume transport of the BSO return-flow and the $SIA_{max}$ time series from each area within the BS. The resulting map of correlation coefficients exhibits the highest values in the central BS, far downstream of the BSO and also clearly downstream of 35°E (Fig. 5), which defines the eastern extent of the AW recirculation cell (Fig. 4a, b). Thus, the correlation is strongest downstream of the recirculation, where either more or less AW is present, depending on the recirculation strength as indicated by the anomalous upper ocean heat content (Fig. 4c, d). It is also located in an area where the winter mean sea ice edge typically resides. Indeed, in the central BS, the sea ice edge during annual maximum reveals pronounced variability coinciding with the area of maximum correlations (Fig. 5). In the western BS, the ice edge is rather stable and spatially confined (Fig. 5). Taking the two aspects (AW circulation variability and sea ice edge position) together, we interpret this as being evidence for the role of the strength of the recirculation of AW—for which the strength of transport of the BSO return-flow is a good indicator—on $SIA_{max}$ in the BS.

### Discussion

In this study, we propose that the strength of recirculation of AW in the BS supplying the BSO return-flow during winter plays a role in governing the interannual $SIA_{max}$ variability in the BS. To the best of our knowledge, this connection has not yet been identified, nor accounted for, in previous studies[3,4,30]. We emphasize that previous findings of the impact of both the AW inflow into the BS and locally wind-driven sea ice drift on $SIA_{max}$ variability are still particularly relevant. The aim of this study is to highlight a previously overlooked process of, in our view, first-order importance. It can be summarized as follows: during winters in which a decrease (increase) in the volume transport of the BSO return-flow is observed, the throughflow of AW towards BSX further to the east strengthens (weakens) as there is almost a full, instantaneous compensation of volume transport anomalies between the return-flow and the flow through BSX. This means that an anomalously weak (strong) recirculation promotes anomalously large (small) amounts of heat to be transported from BSO to the central and eastern BS. Thus, in those years, more (less) ocean heat reaches the sea ice, and consequently, the ocean's role in suppressing (promoting) wintertime ice formation strengthens: the maximum sea ice area in these winters is thus reduced (increased).

In addition, the simulation suggests a major weakening of the return-flow over the 1979–2019 period of 0.76 Sv (representing 60% of the long-term mean value), while there is no such trend in the AW inflow strength in BSO (Fig. 2). The reduction of AW export from the BS into the Nordic Seas, in addition to the rising AW inflow temperatures, may have contributed to the increase in BS heat content and pronounced sea ice retreat. Due to a lack of long-term observations of the strength of the return-flow, the results in this study rely on a model hindcast simulation. Previous evaluations[29], Fig. 1, and Fig. 2 of this study show that this simulation well represents the interannual ocean and sea ice variability in the BS. Most notably, the favorable agreement of the observed temperature time series in the BS return-flow in winter, specifically compiled for this study (Fig. 1b), should give confidence in the robustness of the results presented.

Based on the simulation, we suggest that the interannual variability of the return-flow in the BSO is of comparable magnitude to the variability of the AW inflow, thus representing an important contribution of net AW transport variability in the BS. The simulation generally shows a non-significant correlation between the transport of the AW inflow and that of the return-flow, whereas their temperature

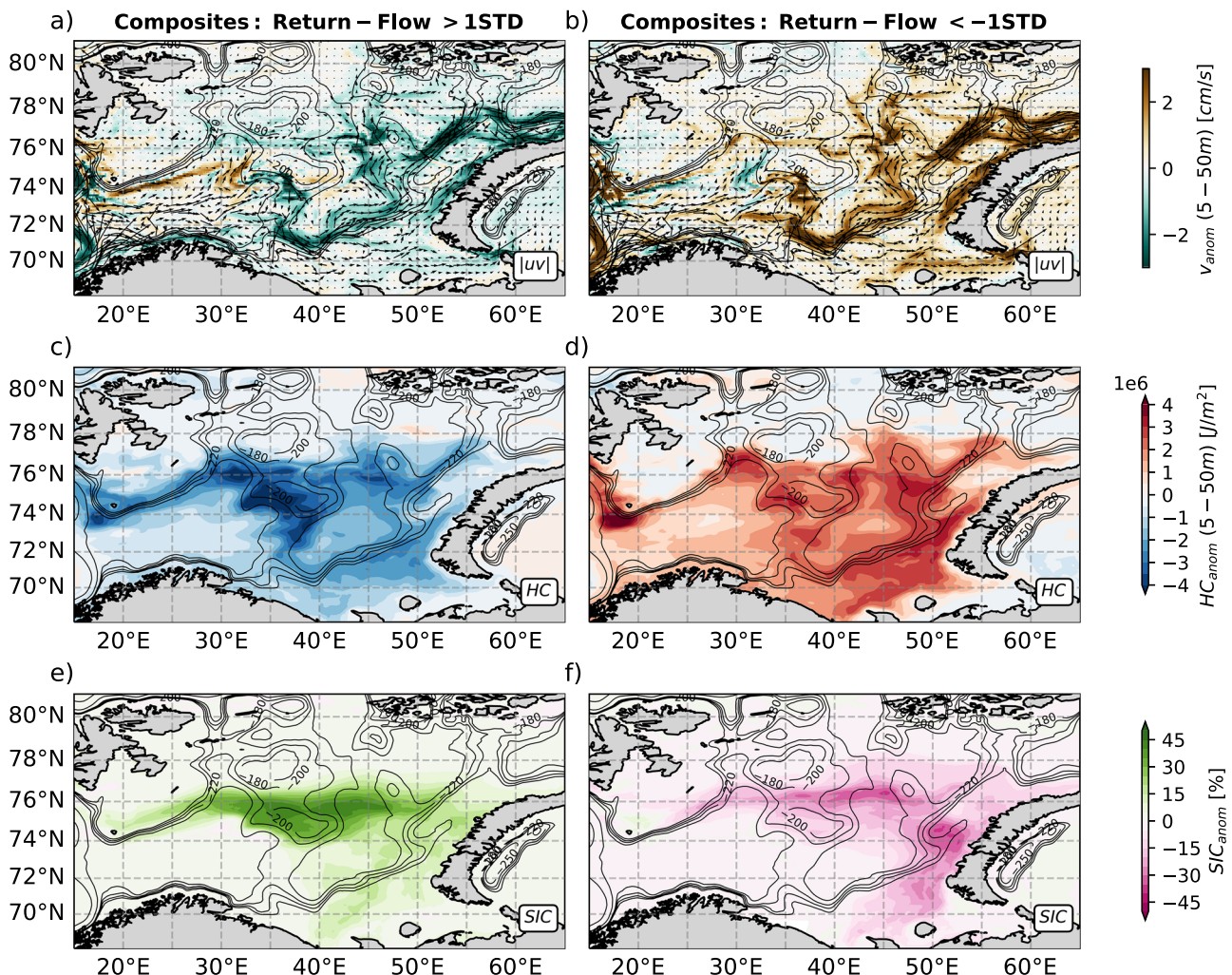

**Fig. 4 | Upper ocean conditions in the Barents Sea during anomalous return-flow.** Composites of anomalous upper ocean absolute velocity (DJFM) in winters with increased (**a**) and decreased (**b**) maximum return-flow. Black arrows depict the upper ocean winter mean velocity. Composites of anomalous upper ocean heat content (HC) in winters (DJFM) with increased (**c**) and decreased (**d**) maximum return-flow. Composites of anomalous sea ice concentration (SIC) in winters (DJFM) with increased (**e**) and decreased (**f**) maximum return-flow. Black contours indicate bathymetry. The threshold for increased/decreased return-flow transport is 1 standard deviation.

variability is strongly correlated. This is in line with results from the 2-year mooring deployment in the return-flow area[27]. The uncorrelated transports support separate and independent (atmospheric) forcing mechanisms driving the volume transport variability of the two branches[7,28,29]. The strong temperature correlation suggests a short recirculation route from inflow to outflow. Whereas the variability of the AW inflow is largely bound to the wind pattern of the North Atlantic Oscillation[9,29], the return-flow appears to be strongly affected by local cyclone activity over Svalbard[7,28,29]. The results obtained in our study may serve as a starting point to identify the processes involved in setting the time-variable strength of the AW recirculation/BSO return-flow. We identified coherent, quasi-simultaneous flow anomalies along the AW pathways through the BS in winter, which ultimately affect the SIA, supporting previous work on short-term AW inflow effects on sea ice[10]. We consider it conceivable that changes in the return-flow driven by local wind anomalies, such as the intensity and frequency of synoptic-scale cyclones over Svalbard, may spin up or down the BSO return-flow depending on the wind direction. This is mainly supported by the barotropic nature of the velocity anomaly in the BSO return-flow in winters with anomalous SIA$_{max}$ (Fig. 3b), typical of coast-parallel wind setting up an across-shelf sea surface height gradient which drives a geostrophic flow along the shelf edge. It is, however, unclear

how the flow anomaly is communicated from the return-flow to the AW circulation branches within the BS that finally result in the coherent anomalous flow structure we revealed in the BS and up to BSX (Fig. 4a, b). Several studies have suggested the existence of a wind-excited eastward, topography-following passage of fast barotropic waves[31] in the Eurasian part of the Arctic Ocean[32]. It could thus be that an initial barotropic adjustment of the return-flow by wind-driven changes of the sea surface height excites barotropic waves, which travel eastward and adjust the AW throughflow. Addressing this question in more detail seems a promising subject of future study. In this regard, the available temporal resolution of the model output in this study, consisting of monthly means, is not nearly sufficient to infer a meaningful lead-lag relation, as the adjustment timescale should be on the order of days rather than months. Suitable observational data would also be necessary, such as tide gauges or satellite-based altimetric observations.

The co-variability of the AW inflow branches into Fram Strait and BSO has been noted previously[28] and further linked to shifts in the AW recirculation branches in Fram Strait[7,33]. Both the strength and the pathways of the AW recirculation in Fram Strait have been found to drive the subsurface supply of warm, saline Atlantic Intermediate Water to the East Greenland shelf[33], thereby controlling the basal melt

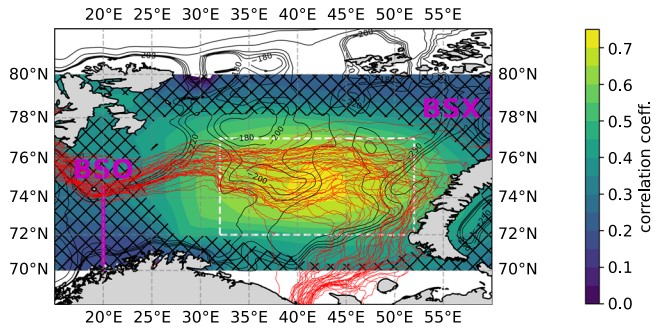

**Fig. 5 | Correlations between sea ice area and return-flow.** Map of Pearson correlation coefficients derived from correlating the Barents Sea Opening (BSO) return-flow with the annual maximum sea ice area in spatial subsets of the Barents Sea, such as the white box. Hatching indicates correlation coefficients that are not statistically significant (99% confidence). Red lines depict sea ice edge (15% sea ice concentration) during annual maximum sea ice area in the Barents Sea from 1979 to 2019. Black contour lines depict bathymetry. The BSO and the Barents Sea Exit (BSX) are denoted as magenta lines.

rates of major outlet glaciers such as the 79N Glacier[34]. The return-flow of AW in the BSO feeds modified (cooled) AW into the AW inflow branch in Fram Strait and, we assume, mostly into the onshore branch of the West Spitsbergen Current which transports the waters along the continental slope of Svalbard where AW subducts to serve as a major heat supply of the Arctic Ocean. As the strength of the BSO return-flow in our simulations has declined over the past decades, this should constitute a weakening element of the inner branch of the WSC. Given that the waters supplied by the BS return-flow should be colder than those in the WSC, the decline of the BS return-flow may have resulted in a temperature increase of the WSC. Assessing the downstream effect of the decline of the return-flow on sea ice decline, not only in BS but also in the Eurasian Basin of the Arctic Ocean associated with the Atlantification[35] remains another fascinating study subject for the future.

Our results also raise the question of whether the BSO return-flow, which is only about 40 km wide, can be adequately represented in the still relatively coarse-resolution ocean models of the Coupled Model Intercomparison Project (CMIP) and whether its postulated influence on the sea ice is thus included in the projections. However, the comparison with the 1/12° (~8 km in BSO) ocean reanalysis GLORYS shows that this resolution could already be sufficient to simulate a similar variability and long-term change in the BSO return-flow. However, an analysis of the CMIP simulations in this respect would be an interesting future research question.

## Methods

### Ocean and sea ice model
The model simulation used in this study was carried out with the Finite volumE Sea Ice and Ocean Model (FESOM2.1), which solves the hydrostatic primitive equations under the Boussinesq approximation with the finite volume method[36–38]. FESOM2.1 is formulated on an unstructured triangular mesh, allowing regional high-resolution mesh refinements in a global coarse-resolution setup. The sea ice component of FESOM2.1 is based on an Elastic-Viscous-Plastic rheology and is solved on the same computational mesh. Here, a global mesh with ~4.5 km grid spacing in the whole Arctic Ocean, as well as in the Nordic Seas, is used. Elsewhere, the resolution is set to nominal 1°. In the vertical, the model uses z coordinates. In total, 46 layers with 10 m layer thickness close to the surface, increasing to 250 m in the deep ocean, are used. Vertical mixing was parameterized by a turbulent kinetic energy scheme computed by the CVMix package[38]. In regions with coarse mesh resolution (>30 km mesh resolution), isoneutral tracer diffusion[39] and the Gent-

McWilliams[40] eddy parameterization are applied. The model was initialized with temperature and salinity fields from the PHC3 climatology[41]. It was forced with the JRA55-do atmosphere reanalysis for driving ocean sea ice models[42]. After a full 1958–2019 model simulation is performed as a spin-up, the model is restarted from the final 2019 conditions for a second full cycle. The results shown in this study are based on the 1979–2019 period of the second cycle. Due to limitations of the computing center, the simulation could not be continued to the present day. The model adequately reproduces both interannual variability and trend of the temperature of the central AW inflow and the return-flow when compared with in-situ observations as well as SIA of the BS (Fig. 1c, d, e).

### GLORYS ocean reanalysis
The GLORYS12V1 product used in this study is the Copernicus Marine Services global ocean eddy-resolving (1/12° horizontal resolution, 50 vertical levels) reanalysis covering the altimetry time period (1993 onward). The ocean model component is the Nucleus for European Modeling of the Ocean (NEMO) platform driven at surface by the European Center for Medium-Range Weather Forecast (ECMWF) ERA-Interim[43] and then ERA5[44] reanalysis for recent years. Observations are assimilated by means of a reduced-order Kalman filter. Along-track altimeter data (Sea Level Anomaly), Satellite Sea Surface Temperature, Sea Ice Concentration, and in situ Temperature and Salinity vertical Profiles are jointly assimilated.

### CTD and moored data
The Institute of Marine Research (IMR, Bergen, Norway) maintains a routine hydrographic section with 20 CTD stations crossing the BSO that has typically been occupied 5–6 times a year since 1977. The five northernmost stations (north of 73.6°N) can be considered to be within the return-flow region. A time series of temperatures within the return-flow was estimated by averaging the observed temperatures at these five stations below 50 m depth. Additionally, a mooring array consisting of (mostly) five moorings within the BSO has been maintained more or less continuously (albeit with some modifications) since 1997[45].

### Volume transport calculation
To compute volume transports on the FESOM2.1 mesh, we use the Python toolbox pyfesom2. In order to separate return-flow and AW inflow, we sum up all grid cells of the BSO (20°E, 70–74.5°N) depth-latitude section that have a westward transport ($u < 0$ ms$^{-1}$) for the return-flow and those that have an eastward transport ($u > 0$ ms$^{-1}$) for the AW inflow. For the calculation of the transports of NCC and central AW inflow, we use a spatial definition (NCC: 20°E, 70.1–71.45°N; central AW inflow: 20°E, 71.45–73.48°N).

### Annual maximum sea ice area
Sea ice area of the BS is computed for every month in the 1979–2019 period. It is computed as the total area bounded by 18°E, 60°E, 68°N, 81°N in which the sea ice concentration exceeds 15%, both for observational data (NSIDC, ERA5) and the model (FESOM2.1). Note that the month of annual maximum sea ice area is different for different years.

### Linear regression
In this study, linear least squares regression is applied between the eastward velocity in the BSO/BSX and the SIA$_{max}$ of the BS. Before executing the respective regression fit, all time-series are linearly detrended in time-space and the mean is removed. The time-series of SIA$_{max}$ is standardized by dividing the detrended anomalous SIA$_{max}$ by its standard deviation. The statistical significance of the regression slopes is tested on a 99% confidence level based on a two-sided hypothesis test.

## Pearson correlation

Pearson correlations are computed between linearly detrended anomalies of the respective quantities. The statistical significance of the correlation coefficients is tested at a 99% confidence level based on a two-sided hypothesis test.

## Composite analysis

For the composite analysis (Fig. 4a, b), only winters where the linearly detrended anomalies of the return-flow volume transport exceed ± 1 standard deviation are used. Before averaging the respective winters, all data were linearly detrended, and the mean was removed.

## Heat content anomaly

Maps of upper ocean heat content anomalies are computed by

$$\Delta HC_i = \rho c_p \cdot \int_{-50m}^{-5m} \Delta T_i(t,z)dz \qquad (1)$$

where $\Delta HC_i$ is the depth-integrated anomaly of the heat content at the $i$th grid cell at a time $t$, $\rho = 1028\,\text{kg m}^{-3}$ is the density of sea water, $c_p = 4190\,\text{J(kg K)}^{-1}$, and $\Delta T_i(t,z) = T_i(t,z) - \overline{T_i}(z)$ is the temperature anomaly at the $i$th grid cell at a time $t$ at a depth level z. $\overline{T_i}(z)$ is the time mean temperature at the $i$th grid cell at a depth level z. Before calculation of $\Delta T_i(t, z)$, the temperature time series is linearly detrended.

## Local sea ice area subsets of the Barents Sea

To narrow down the region in which the $SIA_{max}$ is affected by the return-flow, various locally confined time series of $SIA_{max}$ in the BS are computed. For this, boxes of 20° longitudinal extent and 5° latitudinal extent are chosen, centered at each available grid cell in the BS. For each box, the time series of the annual $SIA_{max}$ is computed. The linearly detrended interannual anomalies of these time series are then Pearson correlated with the linearly detrended volume transport anomalies of the return-flow to localize the region where the variability of the return-flow can best explain the variability of the $SIA_{max}$. The choice of box size stems from a trade-off between a preferably small box size, increasing the spatial precision of the analysis, and avoiding fully ice- or ocean-covered boxes, leading to unwanted plateaus in the computed $SIA_{max}$ time-series.

## Data availability

All data which is displayed in the figures have been deposited in the zenodo database under accession code https://doi.org/10.5281/zenodo.15023365. Due to the large size of the FESOM2 model data, the raw FESOM2 data can not be stored in an online repository and is available on request from the corresponding author. GLORYS12V1 data can be accessed at https://doi.org/10.48670/moi-00021. BSO temperature data are available at https://ocean.ices.dk/core/iroc. ERA5 data can be accessed at https://doi.org/10.24381/cds.f17050d7. JRA55-do-v1.4 is stored at https://climate.mri-jma.go.jp/pub/ocean/JRA55-do/. NSIDC sea ice concentration is available at https://nsidc.org/data/g10010.

## Code availability

The source code of FESOM2.1 is freely available at https://github.com/FESOM/fesom2/releases/tag/2.1.1. The model source code and the model namelists to reproduce the simulations presented are further available in the zenodo database under accession code https://doi.org/10.5281/zenodo.15023365. Pyfesom2 can be downloaded at https://github.com/FESOM/pyfesom2/tree/main.

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

## Acknowledgements

F.O.H. and T.K. gratefully acknowledge the funding by the Deutsche Forschungsgemeinschaft (DFG, German Research Foundation) through the Transregional Collaborative Research Centre TRR-172 "ArctiC Amplification: Climate Relevant Atmospheric and SurfaCe Processes, and Feedback Mechanisms (AC)3" (grant 268020496). C.W. and T.K. gratefully acknowledge funding by the Federal Ministry of Education and Research in Germany (BMBF) through the research program GROCE2 (FKZ 03F0855A). T.M.B., F.O.H., T.K., and R.M. gratefully acknowledge funding by the European Union as part of the EPOC project (Explaining and Predicting the Ocean Conveyor; grant number: 101059547). Views and opinions expressed are, however, those of the author(s) only and do not necessarily reflect those of the European Union. Neither the European Union nor the granting authority can be held responsible for them. We further gratefully acknowledge the computing time granted by the Resource Allocation Board and provided on the supercomputer Lise at NHR@ZIB as part of the NHR infrastructure. The calculations for this research were conducted with computing resources under the project hbk00087. We acknowledge support by the Open Access Publication Funds of Alfred-Wegener-Institut Helmholtz– Zentrum für Polar- und Meeresforschung.

## Author contributions

The study was designed, executed, and written by F.O.H., with C.W., T.K., and R.M. providing valuable ideas, guidance, and engaging in discussions that shaped its development. T.M.B. contributed to the study by creating the observation-based temperature time series in the BSO return-flow. All co-authors contributed to the finalization and improvement of the manuscript.

## Funding

## Competing interests

The authors declare no competing interests.
