## [Transparent Peer Review file · Nature Communications]

Atlantic Water Recirculation in the Northern Barents Sea Affects Winter Sea Ice Extent

Corresponding Author: Dr Finn Heukamp

Version 0:

Reviewer comments:

Reviewer #1

(Remarks to the Author)

Overall recommendation: Major revisions

This manuscript examines "Declining Atlantic Water Return-Flow Suppresses Wintertime Sea Ice Formation in the Barents Sea". This topic is very interesting and meaningful. The author has provided a new perspective to understand what cause Arctic sea ice has declined in all seasons. I thought that this manuscript has potential to be published in Nature communications after provide more enough pieces of evidence in detail. Thus, I recommend that the manuscript needs major revision.

[Editorial Note: Reviewer #1's full report is attached at the end of this file]

Reviewer #2

(Remarks to the Author)

Review: Declining Atlantic Water Return-Flow Suppresses Wintertime Sea Ice Formation in the Barents Sea, by Heukamp et al.

Using a global ocean and sea ice model and observations, this study suggests that the Barents Sea return flow plays an important role in the region's winter sea ice. Their results show that weaker return flow leads to reduced recirculation of Atlantic water, which increases heat in the Barents Sea and suppresses winter ice formation.

This is a promising study with interesting results, but I think it needs a major revision, particularly in providing additional analyses to better support the conclusions and meet the journal's level. Please see my major and minor comments below. If any comment is unclear, please feel free to contact me.

Major comments:

1. In general, I find that some of the main statements are too qualitative for an article of this nature, with several declarative statements made about the results lacking sufficient justification in the text or figures. For instance, one of the primary findings is that the weakening of the return flow increases heat in the Barents Sea (e.g., lines 8-10); however, the paper does not provide any analysis quantifying this heat increase. This can be easily quantified by showing e.g., the heat content and heat fluxes. Another example is the impact on the spatial distribution of sea ice concentration in the Barents Sea. These are just a few instances, and I will elaborate on them in my following comments. Also, better distinction between trend and anomaly should be given when interpreting results.

2. A clear description of the Barents return flow, including what drives it? In Figure SI2 you show the related sea level pressure field, but this should be driving the barotropic component, right? Overall, more can be said about the return flow and its drivers right at the beginning of the paper to make the results of the study clearer.

3. needed analysis:

- Heat transport (time series) for AW and return flow.
- volume transport for the AW and return flow (you only show anomalies).
- similar to figure SI2, but for the trend
- similar to Figure 4, but composites for SST (or heat content e.g. top 50 m) and sea ice concentration
- not sure if the model provides sea ice bottom melt?
- Better distinction between trend and anomaly in results
- include observations in SI1 plot
- sea ice concentration trend: model vs observation

4. For the discussion:

- Please include a subsection discussing the implications of your results for future climate model projections. I am not sure how this overflow is represented in the CMIP models, but my guess is that it is not well captured. Do models with better representation of this return flow show better agreement with sea ice retreat over the observed period?
- We know both temperature and velocity are important when talking about heat transport (Docquier, D., et al. 2021). Could you comment which one is more dominating?

Detailed comments, including minor comments:

Title: "Atlantic water return flow" is misleading. Shouldn't be Barents return flow?

line 4: "... the Atlantic water ...": add (AW)

line 15: remove "central"

line 20-25: needs to be revised. In item (ii) you meant wind-driven? And apart from the mentioned 3 factors, what about atmospheric heat transport and sea ice inflow from north-east?

Line 27: add this after "...unclear": this requires exploring individual modes of variability (cf. Karami et al. 2023)

line 29-30: replace "...lower to high..." with → Atlantic to the Arctic, or something like that

line 33-34: remove "modified (i.e.)"

line 36: first add a sentence why to study AW at BSO. Then this should be developed into describing what Barents return flow is. Highlight the return flow better in the figure.

Line 44-62: Should be a separate sub-section for the return flow and AW (see my comment 2 above)

line 71: remove "For this study"

line 72-77: long sentence; please rewrite

line 80-81: I guess AW inflow should be 2.3? numbers don't match those in the figure

line 85-101: so unclear and poorly written. Why is it interesting? What is the main message here?

Line 104: something is wrong with the sentence

Line 111-115: see my comment 1. It is not clear how you conclude weakened return flow causes sea ice decline. High correlation does not necessarily mean one drive the other.

Line 116-122: Please specify correlation for both full and de-trended time series

line 130: change "our simulation" to → our regression analysis

line 130: "barotropic acceleration"? Revise, e.g., increased barotropic flow...

line 134-138: recalling my comment 2 regarding drivers of return flow.

Line 145-146: remove "This compensation is not provided by ... correlation."

line 155: "decelerated"?

Line 151-163: It is difficult to confirm these conclusions with regression analysis alone (my comment 1). Since you have the velocity and transport fields, you could check the barotropic components of velocity/transport by simply taking the depth-integrated mean, and then continue from there.

Line 186-203: Wouldn't it be better to plot the composite map of sea ice concentration?

References:

Docquier, D., et al. *Clim Dyn* 56, 1407–1432 (2021). <https://doi.org/10.1007/s00382-020-05540-8>

Karami et al 2023 *Environ. Res.: Climate* 2 025005. DOI 10.1088/2752-5295/accbe3

Reviewer #3

(Remarks to the Author)

In the reviewed manuscript the authors describe a new mechanism, contributing to the interannual and long-term variability of winter sea ice area in the Barents Sea. While key drivers of the declining Barents Sea ice cover have been widely discussed in the last decade, the role of Atlantic water recirculation and its return flow in the Barents Sea has not been yet addressed in the context of sea ice changes. A causal link between the strength of the Barents Sea return flow, anomalous circulation and throughflow of Atlantic water, and varying winter sea ice area has been revealed by presented analysis and can be considered as a novel and noteworthy result. Building on this finding, the authors show that a decline of the return flow in the last four decades, as reproduced by their ocean and sea ice model, may have been equally important for the observed decrease of sea ice area in the Barents Sea as previously established mechanisms.

Since the Barents Sea is a hotspot region for ongoing Atlantification of the Arctic Ocean and plays a critical climatic role ("cooling machine") through transformation of water masses and air-sea heat fluxes, the improved understanding of different drivers responsible for shrinking sea ice cover has wide-reaching significance for ocean, climate and ecosystem studies. The Barents Sea return flow of Atlantic water has been only occasionally addressed in the established literature and focused mostly on its qualitative and quantitative assessment or forcing mechanisms. Its role in modifying the amount and pathways of oceanic heat transported through and lost in the Barents Sea as well as the impact on winter sea ice area has not been yet examined. In this aspect, the submitted paper is highly original work.

The main results and conclusions are well supported by the presented work. To explain the mechanisms linking the Barents Sea return flow to sea ice changes, the authors use a state-of-the-art numerical model, validated against available (yet limited) observations. Good model performance in the Barents Sea gives credence to further analysis, based on a hindcast simulation. The methodology applied in the study is sound and adequate for addressing the important yet severely under-observed feature and its large-scale consequences. The data analysis sequence is logical and purposefully structured, the reasoning is coherent and supported by the results, and conclusions are well justified by obtained evidence. Minor flaws or a lack of clarity can be occasionally found in the description of work done or discussion of the results (as indicated in the minor comments below) but they do not diminish the overall value of this study. All figures are relevant and illustrative with extensive captions, providing necessary details. The methods are mostly described with enough detail albeit some information is missing or unclear, e.g. for linear regression and composite analysis. The work presented in the paper can be reproduced when the model outcome becomes available.

Summarizing, the submitted manuscript presents a unique and soundly performed study with results that are of highly relevant for better understanding the ongoing changes in the Barents Sea and their impact on climate and ecosystem. Publication in the *Nature Communications* journal can be recommended after a minor revision, addressing my comments as listed below.

Detailed comments:

Title and line 10: '...suppresses winter ice formation'

The title is not precise as it only refers to the effect of oceanic heat in suppressing winter ice formation. An increased oceanic heat results both in enhanced melting of sea ice (locally formed or advected) and suppressed ice formation in winter. The results described in the paper reflect a decline of sea ice area through both mechanisms. Winter ice formation (as a process) is mentioned only once in the paper what does not justify including it in the title.

Line 8: '...reduced sea ice extent...'

I suggest 'sea ice area' since this parameter is analyzed in the study, not sea extent.

Line 9: '...increased heat in the Barents Sea...'

I suggest 'increased heat content in the Barents Sea' for clarity.

Lines 6, 8, 11: '...Barents Sea return flow...'

In the abstract the name 'return flow' is used while it becomes 'return-flow' through the rest of the paper. Please unify.

Line 20: '...mainly the warming of the inflowing water'

I suggest 'which is mainly due to the warming of the inflowing water.' as it refers to the cause of the increased ocean heat transport.

Line 31: '...where warm and salty Atlantic Water (AW) enters it from the west.'

I suggest 'enter from the west.'

Lines 34-35 and the following sections: '...modified (i.e. cooled and freshened) AW is exported back by a return flow.'
While the whole study refers to Atlantic water in different flows in the Barents Sea (inflows, outflow, throughflow), there is no definition of Atlantic water in terms of water mass. It is not clear whether the Atlantic inflow in the BSO is represented by a total inflow (eastward transport) or an inflow of water defined (how?) as Atlantic-origin water. In the return flow, the authors indicate that Atlantic water constitutes about 80% but do not reveal how is this Atlantic water distinguished from other water masses (in particular after being cooled and freshened during its passage in the Barents Sea). Composite maps (Fig. 4) show, according to the caption, anomalous transport of Atlantic water through the Barents Sea but in reality, AW is not distinguished from other water masses, possibly present at 50 m. I strongly suggest including a definition of Atlantic water either in the introduction or in the methods.

Line 51-52: '...the winter volume transport... .. estimated as 1.6 ± 0.25 Sv, which appears high...'
Observational estimates are in lines 44-45 are given for the annual means while an estimate based on a numerical simulation is calculated as a winter mean. While the argument about winter intensification is valid, a comparison between total annual means of AW transports from observations and model results would be more helpful. It would be also interesting to see how the 2-year averaged volume transport in the return flow obtained from observations compares to simulated 2-year average.

Lines 55-56: '...which is yet unaccounted for in BSO AW (heat) transport estimates.'
This part of the sentence is unclear. Do the authors mean that the return flow may significantly contribute to the variability of AW transport and thus to the variability of heat transport or something else?

Lines 65-66: FESOM abbreviation should be explained when it appears for the first time (i.e., line 51).

Line 74: '...AW temperature variability in the BSO inflow and return-flow...'
Could the authors provide R (correlation coefficient) between observed and simulated time series also for AW temperature variability in the inflow and return flow (as it is done for sea ice area)? Since all three time series are included in Fig. 1, it would give better measure how well does the model reproduce AW temperature in the BSO.

Lines 75-76: '...the volume transport variability of the AW-carrying currents...'
This is unnecessary complication, I suggest 'the variability of AW volume transport...'

Line 87: '...simulation based on the 1979-2019 period'
It is a crucial information about the simulation period and it should be mentioned earlier when the numerical model is introduced. For the estimated AW transports in different branches based on observations, the averaging periods should be also given.

Lines 89-90: '...the mean inflow and return-flow temperatures differ...'
Are these the mean temperatures of entire inflow/return flow or the mean temperatures of AW in the inflow/return flow?

Line 97: 'With AW representing the dominant water mass...'
Again, how is AW defined in the return flow? Does its definition account for cooling and freshening along the route in the Barents Sea?

Line 107: '...of the same freezing season'
It is unnecessary repetition since the season has been already defined (December-March).
Lines 116-122: Correlations between SIAMax and AW volume transport in different branches.
Since it has been well established that amount (variability) of ocean heat carried into the Barents Sea with Atlantic inflow is mostly driven by changes in AW temperature, not volume transport, it would be interesting to see whether also AW temperature in the return flow reflects SIAMax variability, i.e. if stronger cooling of the returning AW is correlated with an increased sea ice loss (due to more ocean heat lost to the ice).

Lines 130-131: '...suggests an overall barotropic acceleration of the return-flow in winters with increased SIAMax'
Since velocity is taken as predictor, the sentence would be more logical when formulated 'suggests increased SIAMax in winters with an overall barotropic acceleration of the return flow'.

Line 183: 'In years with increased SIAMax...'
It rather should be 'decreased SIAMax' when stronger transport in the central Barents Sea provides more heat to the sea ice for melting or inhibiting ice formation.

Lines 195-197: 'This means the correlation is strongest downstream of the recirculation where the variability of the AW circulation should be strongest.'
The meaning of this sentence is unclear, in particular its second part. Why the variability of AW circulations should be strongest (what does it mean - changing pathways, changing strength) downstream of its recirculation cell? If the AW recirculation changes, it implies changes in the downstream AW flow patterns but why should they be stronger than changes in recirculation? Also a statement in lines 200-203 about AW circulation variability and sea ice edge position is a bit obscure.

Line 206: '...plays a major role...'
In my opinion, a statement about 'a major role' is slightly overestimated since there is no objective comparison which driver

of the sea ice area variability is a dominant one. I agree that the strength of AW recirculation is an important driver but not a 'major' one as it has not been compared to other forcings.

Line 215-216: '...there is a near-compensation...'

What is a 'near-compensation'? Do the authors mean 'a near-surface compensation' or the fact that there is nearly full compensation between the return flow and the BSX outflow?

Line 220-221: '...simulation over the 1979-2020 period...'

In the preceding text, the simulation period is defined as 1979-2019.

Line 222: 'The complementary reduction...'

Why 'complementary' when it is entirely different process than the increase of AW temperature in the inflow?

Line 254-255: '...how the flow anomaly is communicated upstream from the Svalbard shelf edge to the AW circulation branches in the BS...'

This sentence is unclear. Do the author suggest that barotropic anomaly of the return flow velocity has its origin on the Svalbard shelf edge? What would be its linkage to the northern BSO?

Lines 318-321: Annual Maximum Sea Ice Area

It should be mentioned that SIAMax is taken for the month with maximum ice area which can be different for different years.

Lines 322-327: Linear regression

Taking into account that on Fig. 4 regression coefficients are expressed in velocity units per standard deviation of SIAMax, all time series are not only detrended and with their means removed but also standardized in some way. This should be mentioned in the description.

Line 333-335: Composite analysis

The parameter which has been used in composite analysis shown on Fig. 4a,b should be indicated in the description. Of three parameters shown on the maps, two are actually winter means, not composites, and only one - anomaly of absolute velocity (is it absolute velocity or rather current speed?) - is a composite for extreme SIAMax years.

Figure 3: The meaning of dotted areas should be explained in the caption.

Figure 4 a and 4 b: Colored isotherms overlaid on color shading for velocity anomaly are hardly legible in areas with stronger anomalies and make it difficult to distinguish the current vectors. In result, the plots are overcrowded even if the spatial distribution of temperature at 50 m is nowhere discussed in the paper. I would suggest removing mean temperature contours from the maps.

Line 471: The reference to Ingvaldsen (2020) lacks a proper DOI. The full citation is:

Randi Ingvaldsen (2020) Mooring data from the Barents Sea Opening – Atlantic Water inflow
<https://doi.org/10.21335/NMDC-1838527821>

Version 1:

Reviewer comments:

Reviewer #1

(Remarks to the Author)

I thank the authors for taking on board my comments which I feel have greatly improved the manuscript, then I agree this manuscript to publish the Nature Communication.

Reviewer #2

(Remarks to the Author)

I don't have further comments, and my major comments were addressed in the review.

Dear Reviewers

We are grateful for your thorough and constructive feedback on our manuscript. Your comments and suggestions have helped us improve the quality and clarity of our work. We appreciate the time and effort you have dedicated to providing detailed assessments, guiding us in addressing key points, and refining our analysis. In the following sections, we have addressed each of your comments in detail and made the necessary revisions to the manuscript. We hope the changes meet your expectations.

Sincerely,
Finn Heukamp

Review #1

#	Reviewer's Comment	Reply
	 Author highlighted that the oceanic processes can impact on the Barents Sea (BS) sea ice decline by the different time scales. However, some studies also found that the atmospheric processes also influence the BS sea ice interannual variability, for example, the winter Barents-Kara Seas (BKS) sea ice associated with Ural blocking (UB) (Luo et al., 2017) and BKS sea ice variability show large interannual variations and related to the UB with the positive phase of North Atlantic Oscillation (NAO+) under the La Niña events (Luo et al., 2023) (-Although the author cites the article, there is no discussion about the article in the full text). Meanwhile, BS sea ice interannual and decadal variability also related to the North Atlantic atmospheric and oceanic transports (Shi et al., 2024). Arctic sea ice loss may also associated with the Arctic amplification under increasing CO₂ (Dai et al., 2019). So, I recommend 	Thank you for highlighting these studies which are certainly of interest to our work. We have now added statements about a few of the named studies in the introduction to better point out the role of the atmosphere in the context of the Barents Sea and provide some examples of how the atmosphere is indeed acting on the sea ice variability. However, the main focus of the manuscript is to investigate oceanic processes, i.e. the impact of the BSO current system on the sea ice variability. In order to keep this focus at the forefront of our results, the references to the atmospheric processes are provided in a brief way. However, we agree with the reviewer that introducing these processes is a fundamental first step to provide the reader with an overview of all processes known to impact the sea ice in the Barents Sea before focusing on the ocean. We have added the following references to the introduction: 6. Kwang Yul Kim, Ji Young Kim, Jinju Kim, Saerim Yeo, Hanna Na, Benjamin D. Hamlington, and Robert R. Leben. Vertical feedback mechanism of winter arctic amplification and sea ice loss. Scientific Reports 2019 9:1, 9:1–10, 2 2019. ISSN 2045-2322. doi: 10.1038/s41598-018-38109-x. 11. Qi Shu, Qiang Wang, Zhenya Song, and Fangli Qiao. The poleward enhanced arctic ocean cooling machine in a warming climate. Nature Communications 2021 12:1, 12:1–9, 5 2021. ISSN 2041-1723. doi: 10.1038/s41467-021-23321-7.

	that author need to add more discussion and references about the relationship between the atmospheric processes and arctic sea ice loss.	19. Tingting Gong and Dehai Luo. Ural blocking as an amplifier of the arctic sea ice decline in winter. Journal of Climate, 30:2639–2654, 4 2017. ISSN 0894-8755. doi: 10.1175/JCLI-D-16-0548.1. 20. Binhe Luo, Dehai Luo, Yao Ge, Aiguo Dai, Lin Wang, Ian Simmonds, Cunde Xiao, Lixin Wu, and Yao Yao. Origins of barents-kara sea-ice interannual variability modulated by the atlantic pathway of el nin˜o–southern oscillation. Nature Communications 2023 14:1, 14:1–13, 2 2023. 21. Zhongfang Liu, Camille Risi, Francis Codron, Zhimin Jian, Zhongwang Wei, Xiaogang He, Christopher J. Poulsen, Yue Wang, Dong Chen, Wentao Ma, Yanyan Cheng, and Gabriel J. Bowen. Atmospheric forcing dominates winter barents-kara sea ice variability on interannual to decadal time scales. Proceedings of the National Academy of Sciences of the United States of America, 119:e2120770119, 9 2022. 22. Dirk Olonscheck, Thorsten Mauritsen, and Dirk Notz. Arctic sea-ice variability is primarily driven by atmospheric temperature fluctuations. Nature Geoscience 2019 12:6, 12:430–434, 5 2019. ISSN 1752-0908. 23. Mehdi Pasha Karami, Torben Koenigk, and Bruno Tremblay. Variability modes of september arctic sea ice: drivers and their contributions to sea ice trend and extremes. Environmental Research: Climate, 2:025005, 6 2023.
	 Why did author used the NSIDC instead of ERA5 dataset (eg: sea ice extent or sea ice cover)? 	We use NSIDC data instead of ERA5 data for sea ice area since we prefer observations over reanalysis data for evaluating the performance of our FESOM simulation. We, however, have added the respective timeseries of annual maximum Barents Sea sea ice area computed from the ERA5 reanalysis to Fig. 1. It agrees well with both NSIDC data and FESOM.

	 And why did the author choose the time period of 1979-2019 instead of 1979-2022 or 1979-2023? Is it due to the limitation of the length of available sea ice area data or another reason? It is suggested that the author explain this reason why you choose the 1979-2019? (or supplement file with sea ice cover with ERA5 data dataset). 	We use the 1979-2019 period since the examined simulation is the same as in Heukamp et al. 2023 (https://doi.org/10.1038/s43247-023-00985-1). This simulation is not available for the years after 2019. It only spans from 1958 to 2019 of which we chose the satellite era starting in 1979 to obtain a reliable evaluation of the sea ice cover. The HPC system we rely on with this particular simulation was recently updated, making it impossible for us to run the last couple of years from 2019 to the present day with the exact same model setup. Adding the near-present years would be great, however, in our view, these few years which are not available do not reduce the significance of the findings. We added a statement in the Data and Methods section, subsection Ocean and Sea Ice Model: „Due to limitations of the computing center, the simulation could not be continued to the present day.“ (ll. 381-382)
	 Line 65, “Finite VolumE Sea Ice and Ocean Model (FESOM2.1)” changed to “Finite Volume Sea Ice and Ocean Model (FESOM2.1)”. 	“Finite VolumE Sea Ice and Ocean Model (FESOM2.1)” is the official notation which we would like to keep.
	 Line 172, “STD” changed to “standard deviation (STD)?” 	We have now introduced the abbreviation STD for standard deviation at its first occurrence in new line 54.
	 Line 272, “79N” changed to “79oN”. 	Thanks for noting this. This glacier is indeed known as the 79N glacier without a degree symbol.
	 Line 334, Author used the SIAMax exceed ± 1 standard deviation. 	Based on your comment, we re-evaluated our choice of using years with SIAMax exceeding ± 1 STD.

	 • Why author do not used the ± 0.5 standard deviation SIAmax? • Is there any difference in SIAmax and their composite results between ± 0.5 and ± 1 standard deviation? • Please add the composite results with the ± 0.5 standard deviation in supplementary file. 	Since it is a key point of the manuscript that an increased volume transport of the return-flow in winter has an effect on the maximum sea ice area (at the end of the same winter) it makes, in our view, more sense to look at those years where the return-flow is in particular strong/weak and not SIAmax. Since both properties (SIAmax and return-flow strength) are, as noted in Fig. 2c, highly correlated, the resulting composites are indeed very similar. To put things in the right order, i.e. a modified return-flow causing changes in the sea ice, we believe that this makes more sense. We chose ± 1 STD since, when looking at the timeseries of the return-flow, it includes 7 out of 42 years for > 1STD and 6 out of 42 for < -1STD (Fig SI4). To us, this seems an appropriate sample size for computing composites, provides a balance between capturing enough data points for analysis while avoiding excessive noise that could arise from smaller deviations. Values within ± 0.5 STD are closer to the mean and may not sufficiently highlight meaningful deviations or anomalies in the data. By selecting ± 1 STD, it is more likely to represent significant variations beyond typical fluctuations in the return flow The choice is, however, arbitrary. We have added the $\pm \frac{1}{2}$ STD composites as requested to the supporting information. It shows similar, however, weaker patterns. We added respective Figures as Fig. SI4, SI5.
--	---	--

Fig. S14: Volume Transport Anomalies of the BSO Return-Flow. Timeseries of detrended and anomalous volume transport of the BSO return-flow. Dashed and dotted lines indicate ± 1 STD and ± 0.5 STD. Red and blue markers indicate winters which are used in the composite analysis presented in Fig. 4.

		  Composites : Return – Flow > 1/2 STD    Composites : Return – Flow < -1/2 STD      $V_{anom} (5 - 50m) [cm/s]$            $HC_{anom} (5 - 50m) U/m^2$            $SIC_{anom} [%]$  Fig. S15: Anomalous Atlantic Water transport through the Barents Sea. As in Fig. 4, but for years where the anomalous transport of the return-flow exceeds ± 0.5 STD (Fig. S14).
	 The authors used FESOM2.1 model to verify that return-flow determines AW 	Thank you for this suggestion. In order to evaluate whether other models depict a similar link between the variability and trend of the Atlantic Water recirculation in

	volume and heat transport in the BSO region, then cause the SIC decline over the BSO region, Whether other models will have the same results? So I suggest that the authors need to consider other models to validate this results, such as using the MITgcm, CESM and Cmp6 models results, or add more discussion about the models results in discussion part.	the northern BSO and the annual maximum sea ice extent, we turned our attention towards the ocean reanalysis GLORYS12v1, a 1/12° reanalysis based on the NEMO ocean model. We chose GLORYS over another “free” model such as FESOM2.1 as it particularly assimilates satellite altimeter data. This is advantageous as the barotropic transport in GLORYS is strongly affected by observed sea surface height gradients. As shown in Fig. 2b and d, FESOM2.1 and GLORYS transports of the return-flow reveal a strong similarity, therefore supporting the explanatory power of FESOM2.1 and the results presented in this study. We have added according statements throughout the manuscript:  1. “In order to put our model results on a broader basis, especially with regard to the simulated volume transports of the return-flow, which cannot be evaluated with in-situ measurement data, we also make use of the ocean reanalysis GLORYS12V1 (GLORYS), available from 1993 onward. With a horizontal resolution of about 8 km, GLORYS has only half the resolution of FESOM2.1, but the assimilation of in-situ data and especially satellite altimetry data allows an estimation of the performance of the model. For the 1993-2019 period, in which model and reanalysis are both available, the winter mean volume transport of the return-flow in the ocean reanalysis is slightly weaker (model: 0.85 Sv, reanalysis: 0.56 Sv) (Fig. 2b). In terms of interannual variability of the return-flow’s volume transport, both model and reanalysis indeed reveal similar orders of magnitude (standard deviations of the linearly detrended data 1993-2019: model: 0.36 Sv, reanalysis: 0.32 Sv) and are further strongly correlated (R=0.72, p<0.01) (Fig 2b, d). The consistency between the model and reanalysis provides confidence in the model’s ability to simulate the interannual transport variability of the return-flow in the BSO.” (ll. 131-143)
--	---	---

		2. “Our results also raise the question of whether the BSO return-flow, which is only about 40 km wide, can be adequately represented in the still relatively coarse-resolution ocean models of the Coupled Model Intercomparison Project (CMIP) and whether its postulated influence on the sea ice is thus included in the projections. However, the comparison with the 1/12 ° (approx. 8 km in BSO) ocean reanalysis GLORYS shows that this resolution could already be sufficient to simulate a similar variability and long-term change in the BSO return-flow. However, an analysis of the CMIP simulations in this respect would be an interesting future research question.” (ll. 354-361) 3. “The GLORYS12V1 product used in this study is the Copernicus Marine Services global ocean eddy-resolving (1/12 ° horizontal resolution, 50 vertical levels) reanalysis covering the altimetry (1993 onward). The model component is the Nucleus for European Modeling of the Ocean (NEMO) platform driven at surface by European Center for Medium-Range Weather Forecast (ECMWF) ERA-Interim then ERA5 reanalyses for recent years. Observations are assimilated by means of a reduced-order Kalman filter. Along track altimeter data (Sea Level Anomaly), Satellite Sea Surface Temperature, Sea Ice Concentration and In situ Temperature and Salinity vertical Profiles are jointly assimilated. ” (ll. 385-393)
--	--	--

Review #2

#	Reviewer's Comment	Reply
1	A clear description of the Barents return flow, including what drives it? In Figure SI2 you show the related sea level pressure field, but this should be driving the barotropic component, right? Overall, more can be said about the return flow and its drivers right at the beginning of the paper to make the results of the study clearer.	We have added a respective paragraph to the introduction, properly explaining the drivers of the barotropic and baroclinic components of the return-flow in the BSO. The respective paragraph now reads: "In regard of this variability, barotropic transport anomalies of the return-flow are mainly driven by air pressure anomalies over Svalbard^{28,7,29}. Associated (anti-)cyclonic wind anomalies modify the divergence of the Ekman transport onto or off the northern BS/Svalbard shelf. In turn, the meridional sea surface height gradient in the northern BSO which drives the return-flow is modified⁷. In addition, the warm and saline AW south of the return-flow and the cold and fresh Polar Water north of the return-flow result in a strong meridional density gradient across the return-flow, adding a pronounced baroclinic component to the velocity field, which is reflected in a bottom-intensification of the return-flow." (ll. 70-78)
2	Needed analysis: Heat transport (time series) for AW and return flow.	Ocean heat transport estimates in the BSO have been often used. Since the volume transport through BSO is non-zero, the volume-budget is not closed. In such a setting, the estimates of heat transport depend on the units used for temperature (°C vs K), more precisely the reference temperature, which would not be the case for a closed volume budget. Often, arbitrary reference temperatures such as freezing point of sea water or freshwater are used, which however do not solve the problem (see Schauer et al. 2009, https://doi.org/10.5194/os-5-487-2009).

We thus stay away from making heat transport calculations, as the results would be arbitrary by default. The arbitrariness not only affects the mean value but also the magnitude of temporal variability and in case of the BSO also the ratio between heat transport into and out of the Barents Sea.

In Schauer et al. 2009 the arbitrariness of heat transports through partial sections (non-closed budget) is exemplarily shown for the BSO using Celsius and absolute temperature scales:

Partial Section	Transport (Sv)	Temp (°C)	Heat Transport Tref=0°C (TW)	Temp (K)	Heat Transport Tref=0K (TW)
Barents Sea in	2.5	5	50	278	2780
Barents Sea out	-0.5	2	-4	275	-550

It becomes evident that the ratio between the BSO in/out “heat transports” computed using Celsius scale is $50 \text{ TW} / -4 \text{ TW} = -12.5$ and in absolute temperature scale it is $2780 \text{ TW} / -550 \text{ TW} = -5.05$. Hence, we do not provide any comparison between the “heat transport” of the AW inflow and the return flow.

2.1

volume transport for the AW and return flow (you only show anomalies).

We have amended Figure 2 accordingly. It now shows the total AW inflow in winter (a), the total return-flow in winter (b), and the respective anomalies of

the AW inflow and the return-flow in comparison to the SIA.

Figure 2: Atlantic Water Flow and Sea Ice Area Variability in the Barents Sea. (a) Winter volume transport (December-March mean) of the AW inflow and (b) of the return-flow for period 1979-2019 (FESOM2.1) and 1993-2019 (GLORYS). Note that positive numbers in (b) indicate an increased return-flow. Statistically significant linear trends (95% confidence) are shown in (b) as dashed lines. (c) Anomalous and detrended AW inflow, return-flow, and maximum annual BS SIA (cyan bars, right axis) for the period 1979-2019. Dotted magenta lines depict the standard deviation of the anomalous return-flow volume transport. (d) Anomalous and detrended return-flow for FESOM2.1 and GLORYS for the period 1993-2019.

2.2

similar to figure SI2, but for the trend

We have added an according Figure to the supporting information, Fig. SI2.

Fig. SI2: Atmospheric driving pattern of the Atlantic Water return-flow. (a) The pattern of anomalous sea level air pressure from a linear regression analysis with the detrended anomalous volume transport of the return-flow as a predictor of the detrended anomalous sea level air pressure (1979-2019). (b) Linear Trend of the sea level air pressure over the northern Barents Sea (1979-2019). Red dot indicates location of most maximum trend in sea level air pressure. Dotted areas in (a) and (b) indicate non-significant regression coefficients (95% confidence). (c) Timeseries of winter mean sea level air pressure at location of maximum trend (red dot in (b)) with its long-term trend and the 95% confidence interval of the regression slope.

		Indeed, the long-term trend in sea level air pressure (b) shows a pattern particularly similar to the driving pattern of the barotropic transport anomalies of the return-flow (a) and is negative (-1 hPa/decade), supporting that the barotropic component of the return-flow has weakened over the past decades as the atmospheric circulation became “more cyclonic”, weakening the SSH gradient driving the barotropic component of the return-flow. We have modified the manuscript accordingly: „Given the time-mean baroclinic, bottom-intensified structure of the return-flow, the barotropic structure emerging from the regression fit points towards a wind-driven rather than a density-driven modification of the current. This is further supported by a local sea level air pressure anomaly over the north-western BS emerging in winters with pronounced return-flow (Fig. SI2a). The associated anomalous anticyclonic winds drive Ekman transport onto Svalbard Bank, ultimately controlling the sea surface height gradient which determines the barotropic component of the AW return-flow^{29,30}. In this regard, the general weakening of the return-flow coincides with a slight trend in sea level air pressure (approx. -1 hPa/decade) over the Svalbard shelf (Fig. SI2b, c).“ (ll. 189-197)
2.3	similar to Figure 4, but composites for SST or heat content (e.g. upper 50m) and sea ice concentration	Thank you for this suggestion, we have added the respective composites to

Fig. 4.

Figure 4: Upper Ocean Conditions in the Barents Sea during Anomalous Return-Flow. Composites of anomalous upper ocean absolute velocity (DJFM) in winters with increased (a) and decreased (b) maximum return-flow. Black arrows depict the upper ocean winter mean velocity. Composites of anomalous upper ocean heat content (DJFM) in winters with increased (c) and decreased (d) maximum return-flow. Composites of anomalous sea ice concentration in winters with increased (e) and decreased (f) maximum return-flow. Black contours indicate bathymetry.

We further modified the according paragraph in the manuscript. It now reads:

		“The results reveal coherent patterns of flow variability along the AW circulation pathways connecting the BSO and the BSX (Fig. 3b, d; Fig. 4a, b). Specifically, our simulation indicates that in winters with increased return-flow the velocities along both the northern and the more pronounced southern AW pathway through the BS east of roughly 35°E are decelerated, while the recirculation mainly happening west of 35°E and supplying the BSO return-flow with AW is strengthened (Fig 4a). This shows that AW transport towards the central and western BS is reduced as AW is instead redirected out of the BS by the recirculation and the return-flow. As a consequence of these circulation changes, a pronounced negative anomaly of the upper ocean heat content is found in the central and western BS in winters with pronounced return-flow (Fig. 4c). The reduced central BS ocean heat content is further reflected in increased sea ice concentration (Fig. 4e). In years with decreased return-flow all anomalies are of opposite sign (Fig. 4b, d, f), meaning accelerated circulation through the BS (Fig. 4b), increased upper ocean heat content (Fig. 4d) and reduced sea ice concentration (Fig. 4f). Hence, our results indicate that pronounced AW recirculation and return-flow cause an upper ocean in the central BS that is anomalously cold, as a decent fraction of the AW heat is removed from the BS, facilitating formation and persistence of sea ice and thus allowing increased SIA_{max}. In contrast, in weak return-flow years, the AW heat penetrates deep into the BS, facilitating sea ice melt and thus reduces SIA_{max}. ” (ll. 237-254)
2.4	not sure if the model provides sea ice bottom melt?	No, the model does not provide sea ice bottom melt, only the absolute thermodynamic growth rate of sea ice. Since we cannot split the absolute growth rate into ocean- and atmosphere-driven, it is not useful in this case.
2.5	Better distinction between trend and anomaly in results	We have amended the text to better distinguish between trend and variability.

		In the first results section „ Westward Return-Flow in the Northern BSO: A Key Driver of Atlantic Water Transport Variability “ we now solely and explicitly discuss variability before including the trend in the second results section „ A Link between the Atlantic Water Return-Flow in the Barents Sea Opening and Sea Ice Area in the Barents Sea “.
2.6	include observations in SI1 plot	We included the winter means of the temperature calculated from the observations shown in Fig. 1 c, d to the Figure SI1.

Fig. S11: Atlantic Water temperature in the Barents Sea Opening. (a) Winter mean (December-March) (a) AW temperature in the central AW inflow of the BSO at 73°N (50-200m) derived from regular CTD stations and FESOM2.1 and in the return-flow of the BSO between 73.67°N and 74.25°N (50 m - bottom, see methods) derived from regular CTD stations and FESOM2.1. (b) As in (a) but with the long-term trend and the mean removed.

2.7

sea ice concentration trend: model vs observation

Please see comment 26 on sea ice concentration trends. In Fig 1c the integrated SIA trend is shown which is well represented by the model.

3	For the discussion: Please include a subsection discussing the implications of your results for future climate model projections. I am not sure how this overflow is represented in the CMIP models, but my guess is that it is not well captured. Do models with better representation of this return flow show better agreement with sea ice retreat over the observed period? We know both temperature and velocity are important when talking about heat transport (Docquier, D. et al. 2021). Could you comment which one is dominating?	This is indeed a very interesting research question. However, fully evaluating the return-flow CMIP models would need a completely new analysis which is beyond the scope of this study. However, we can compare FESOM with the ocean reanalysis GLORYS12V1 with 1/12° (8km in BSO) resolution, based on the NEMO ocean model, and we find that GLORYS simulates a very similar return-flow when compared to FESOM. The strong similarity in the transports of the return flow are shown in Fig. 2B and d. Therefore it seems that a resolution as high as 4.5km in FESOM may not be needed and 1/12° degree might be sufficient to adequately represent the return-flow. We have added a paragraph at the end of the discussion on this topic. The respective paragraph reads: “Our results also raise the question of whether the BSO return-flow, which is only about 40 km wide, can be adequately represented in the still relatively coarse-resolution ocean models of the Coupled Model Intercomparison Project and whether its postulated influence on the sea ice is thus included in the projections. However, the comparison with the 1/12° (approx. 8 km in BSO) ocean reanalysis GLORYS shows that this resolution could already be sufficient to simulate a similar variability and long-term change in the BSO return-flow. However, an analysis of the CMIP simulations in this respect would be an interesting future research question.” (ll. 354-361) Please see our comment regarding your comment #2 on why we do not use heat transports or any associated parameters.
4	Title: “Atlantic water return flow” is misleading. Shouldn’t be Barents return flow?	We have modified the title. The title now reads: “Atlantic Water Recirculation in the Northern Barents Sea Affects Winter Sea

		Ice Extent
5	line 4: "... the Atlantic water ...": add (AW)	Following the nature communications style guidelines, abbreviations are not allowed in the abstract. We have replaced the abbreviation "AW" in line 12 with the full term "Atlantic Water"
6	line 15: remove "central"	We removed the term "central"
7	line 20-25: needs to be revised. In item (ii) you meant wind-driven? And apart from the mentioned 3 factors, what about atmospheric heat transport and sea ice inflow from north-east?	Yes, we meant wind-driven and have modified the text accordingly. As you suggested, we also added a statement in the Introduction about the role of atmospheric heat transport in driving sea-ice variability and added relevant literature accordingly. Sea ice inflow from the north and the east is already mentioned in the manuscript as a source of sea ice variability, namely: "wind-driven sea ice drift through the eastern and northern gates connecting the BS to the Arctic Ocean and the Kara Sea." The respective paragraph now reads: " The interannual variability of the BS sea ice has, however, been ascribed to variations in  - ocean heat content, driven, with a lag of one year, by the varying ocean heat transport through the western Barents Sea Opening (BSO)^{3,8,9,5,10,11,12,13,14}, - wind-driven sea ice drift through the eastern and northern gates connecting the BS to the Arctic Ocean and the Kara Sea^{4,15,16,17,18}, and - atmospheric processes, such as pronounced Ural blocking¹⁹, La Nina events²⁰, regional anticyclonic wind anomalies²¹, and air temperature fluctuations²².

		However, the relative importance of these oceanic and atmospheric processes and the time scales on which they occur remain unclear²³. Here, an additional oceanic contribution to sea ice variability in the BS is proposed. ” (ll. 29-39)
8	Line 27: add this after “...unclear”: this requires exploring individual modes of variability (cf. Karami et al. 2023)	Thanks for pointing to this publication – we have added the publication as reference ²³ .
9	line 29-30: replace “...lower to high...” with → Atlantic to the Arctic, or something like that	We modified the respective lines accordingly.
10	line 33-34: remove “modified (i.e.”	We modified the respective lines accordingly.
11	line 36: first add a sentence why to study AW at BSO. Then this should be developed into describing what Barents return flow is. Highlight the return flow better in the figure.	We have modified the manuscript accordingly. The paragraph now reads: “ The BS plays a key role in transporting ocean heat from the Atlantic Ocean to the Arctic Ocean^{24,8,9} and is thus of particular interest. It is connected to the Nordic Seas via the BSO, where warm and saline Atlantic Water (AW), which is the dominant source of ocean heat of the BS, enters from the west^{25,26} (Fig. 1a). The inflow of AW into the BS occurs in the southern and central parts of the BSO through the Norwegian Coastal Current (NCC) and the central AW inflow. In the northern part, cooled and freshened AW is exported back into the Nordic Seas by a return-flow²⁷, which is fed by a recirculation of AW in the BS^{27,28,7,29} (Fig. 1a).” (ll. 41-47) We have further highlighted the return flow in Fig. 1.

12	Line 44-62: Should be a separate sub-section for the return flow and AW (see my comment 2 above)	We have better separated the paragraph on the return flow from the paragraph on the AW inflow. As pointed out above, in the first results section „Westward Return-Flow in the Northern BSO: A Key Driver of Atlantic Water Transport Variability“ we now solely and explicitly discuss variability before including the trend in the second results section „A Link between the Atlantic Water Return-Flow in the Barents Sea Opening and Sea Ice Area in the Barents Sea“. We have further amended the introduction according to your comment: “ The AW inflow into the BSO has been monitored by the Norwegian Institute of Marine Research since 1997, which has maintained an array consisting of 5 moorings designed to capture the central AW inflow into the BS, carrying most of the AW heat⁹ across 19.7°E between 71.5°N and 73.5°N^{3,25,26} (Fig 1b). In terms of the volume transport, the NCC carries 1.2 Sv into the BS⁹ and the central AW inflow carries 2.3 Sv into the BS³. The interannual variability (standard deviation (STD) of the annual means) of the central AW inflow is estimated at 0.4 Sv³. The return-flow in the northern BSO is less well documented as mooring deployments there are particularly risky due to intense fishery activity in this region. Nevertheless, the return-flow was monitored over two years when an additional mooring had been deployed at 19.25°E, 73.85°N from September 2003 to October 2005²⁸ (Fig. 1b). Based on this mooring it was estimated that the return-flow carries 0.9 Sv (annual mean) out of the BS toward Fram Strait²⁸ of which approximately 80% represents AW²⁷. No estimates of the return-flow’s interannual variability have, however, been acquired due to the lack of multi-year observations. Based on a numerical simulation, the winter volume transport of the return-flow and its variability has been estimated as 1.6 ± 0.25 Sv²⁹ (1970-2019), which appears high compared to the (limited) observations, but may in part be explained by the generally strong winter
----	--	--

		intensification of the velocity field in the BSO^{25,26,28} and the pronounced downward trend over the entire simulated period²⁹. These previous findings, however, suggest that the return-flow may significantly contribute to the interannual variability of the net AW transport through the BSO which is yet unaccounted for in observational BSO AW transport estimates which only consider the AW inflow regions^{9,24,13}. In regard of this variability, barotropic transport anomalies of the return-flow are mainly driven by air pressure anomalies over Svalbard^{28,7,29}. Associated (anti-)cyclonic wind anomalies modify the divergence of the Ekman transport onto or off the northern BS/Svalbard shelf. In turn, the meridional sea surface height gradient in the northern BSO which drives the return-flow is modified⁷. In addition, the warm and saline AW south of the return-flow and the cold and fresh Polar Water north of the return-flow result in a strong meridional density gradient across the return-flow, adding a pronounced baroclinic component to the velocity field, which is reflected in a bottom-intensification of the return-flow.” (ll. 49-78)
13	line 71: remove “For this study”	We modified the manuscript accordingly.
14	line 72-77: long sentence; please rewrite	We made two parts out of the one sentence.
15	line 80-81: I guess AW inflow should be 2.3? numbers don't match those in the figure	The restructuring of the two paragraphs about the AW inflow and return-flow (as requested above) make the separation between the observational and model estimates of the respective transports clearer. The transports in the Figure are observational estimates which is noted in the caption.
16	line 85-101: so unclear and poorly written. Why is it interesting? What is the main message here?	We have modified large parts of the Introduction. Please look at the revised manuscript for the modified Introduction.
17	Line 104: something is wrong with the sentence	The sentence has been reworded

18	Line 111-115: see my comment 1. It is not clear how you conclude weakened return flow causes sea ice decline. High correlation does not necessarily mean one drive the other.	The reviewer is correct in saying that high correlation doesn't mean causation and it is not our intention to quantify the absolute impact of the return-flow on sea ice in the Barents Sea. However, there is, in our opinion, clear evidence from the model that there is an impact of the return-flow on the sea ice, e.g. in pronounced return-flow years, a substantial part of the AW that enters the BS leaves the BS right away with only limited contact to sea ice. The added heat content anomalies (Fig. 4c, d) give further support to the return-flow affecting the sea ice. In years of strong return flow, the upper ocean in the Barents Sea is cold, as a considerable fraction of the AW heat is removed from the BS by the return-flow, whereas in weak return-flow years the heat penetrates deep into the eastern BS affecting the sea ice. We rather perceive our manuscript as indicating a gap in studies about this region so far that should be addressed as it may improve already established relations.
19	Line 116-122: Please specify correlation for both full and de-trended time series	As the Pearson correlation assumes stationarity (constant mean and variance), computing a Pearson correlation for timeseries with a long-term trend violates this assumption as the trend introduces non-stationarity and can lead to spurious correlation, where the apparent relationship between two variables is driven by the trend rather than any meaningful interaction. Therefore, we only provide the numbers for the detrended timeseries in the manuscript as removing the trend component before computing the correlation provides more reliable insights into the relationships between time series variables.
20	line 130: change "our simulation" to → our regression analysis	We modified the manuscript accordingly.
21	line 130: "barotropic acceleration"? Revise, e.g., increased barotropic flow...	We modified the manuscript accordingly.

22	line 134-138: recalling my comment 2 regarding drivers of return flow.	We have added respective figures to the supporting information. Please review our reply to your respective comment.
23	Line 145-146: remove “This compensation is not provided by ... correlation.”	We do not understand why we should remove the respective sentence.
24	line 155: “decelerated”?	We do not understand what the confusion could be with the term ‘decelerated’ in this context. As shown in Fig. 3d, the eastward flow from the Barents Sea to the Kara Sea is weaker in winters with pronounced SIAMax.
25	Line 151-163: It is difficult to confirm these conclusions with regression analysis alone (my comment 1). Since you have the velocity and transport fields, you could check the barotropic components of velocity/transport by simply taking the depth-integrated mean, and then continue from there.	We do not really understand what you mean here. The regression analysis shows that the velocity anomaly in the BSO is, indeed, barotropic (Fig. 2b). Hence, the timeseries of the return-flow volume transport anomalies that acts as a basis for this study does already represent a barotropic transport anomaly. Moreover, in Fig. 4a, b we show the depth-averaged anomalous velocity fields in the Barents Sea, thus also an estimate of the barotropic velocity field of the upper 50m.
26	Line 186-203: Wouldn't it be better to plot the composite map of sea ice concentration?	In principle, we tend to agree with you. However, we purposely avoided using sea ice concentration for the following reason: In the Barents Sea, the sea ice area reveals a strong downward trend during the last five decades. Sea ice concentration is not a continuous number but is bounded by 0% and 100% and usually takes one of these values (bimodal distribution). Removing the trend from such timeseries that is bounded by maximum/minimum allowed values causes troubles that should be kept in mind. In this regard, please see the example plot (no real data) below that exemplarily shows the issues that arise with linearly detrending data bounded by a given maximum/minimum.

The linear regression line exceeds the maximum allowed value 100% in the first years, leading to the behaviour that the anomaly of the SIC (when detrending and removing the mean) is negative in the first years, then becomes positive and drops back to negative, which clearly does not represent the anomalies of the SIC timeseries. The part of the respective timeseries that can be detrended in a meaningful manner (e.g. year 17-23 in the plot above) is different for each grid-cell.

Thus, to avoid these issues, we chose an integrated property, i.e. the local SIA which is only bounded by 0. We took extra care, that the boxes are large enough that the total ice area in each box never reaches 0 and thus can be detrended without issues.

We agree that providing according composite maps of SIC is, nevertheless, common practice. We thus have added according SIC maps to Fig. 4.

Reviewer #3

#	Reviewer's Comment	Reply
1	Title and line 10: '...suppresses winter ice formation' The title is not precise as it only refers to the effect of oceanic heat in suppressing winter ice formation. An increased oceanic heat results both in enhanced melting of sea ice (locally formed or advected) and suppressed ice formation in winter. The results described in the paper reflect a decline of sea ice area through both mechanisms. Winter ice formation (as a process) is mentioned only once in the paper what does not justify including it in the title.	Thank you very much for pointing this out and we agree with your line of argumentation. We amended the title to be more accurate and in line with this point, and is now titled Atlantic Water Recirculation in the Northern Barents Sea Affects Winter Sea Ice Extent'
2	Line 8: '...reduced sea ice extent...' I suggest 'sea ice area' since this parameter is analyzed in the study, not sea extent.	You are right. We modified the paragraph accordingly.
3	Line 9: '...increased heat in the Barents Sea...' I suggest 'increased heat content in the Barents Sea' for clarity.	We modified the paragraph accordingly.
4	Lines 6, 8, 11: '...Barents Sea return flow...' In the abstract the name 'return flow' is used	We unified the respective appearances of "return flow" to "return-flow" throughout the manuscript.

	while it becomes 'return-flow' through the rest of the paper. Please unify.	
5	Line 20: '...mainly the warming of the inflowing water' I suggest 'which is mainly due to the warming of the inflowing water.' as it refers to the cause of the increased ocean heat transport.	We have modified the respective lines accordingly.
6	Line 31: '...where warm and salty Atlantic Water (AW) enters it from the west.' I suggest 'enter from the west.'	We have modified the respective lines accordingly.
7	Lines 34-35 and the following sections: '...modified (i.e. cooled and freshened) AW is exported back by a return flow.' While the whole study refers to Atlantic water in different flows in the Barents Sea (inflows, outflow, throughflow), there is no definition of Atlantic water in terms of water mass. It is not clear whether the Atlantic inflow in the BSO is represented by a total inflow (eastward transport) or an inflow of water defined (how?) as Atlantic-origin water. In the return flow, the authors indicate that Atlantic water constitutes about 80% but do not reveal how is this Atlantic water distinguished from other water masses (in	The definition of AW has been clarified in two respects. Firstly, in the first paragraph of the results section we now state: “To better understand the role of the return-flow in the net AW volume transport and its interannual variability, we divided the simulated volume transports into components contributing to the eastward flow (NCC and central Atlantic Water inflow) and those contributing to westward flow (return-flow) by separating the flow by its direction (eastwards or westwards).” (ll. 106-114) Additionally, we have also added a more precise description to the methods where we define AW with respect to the transports, and reads: “In order to separate return-flow and AW inflow, we sum up all grid cells of the BSO (20°E, 70-74.5°N) depth-latitude section that have a westward transport ($u < 0$ m/s) for the return-flow and those that have an eastward transport ($u > 0$

	particular after being cooled and freshened during its passage in the Barents Sea). Composite maps (Fig. 4) show, according to the caption, anomalous transport of Atlantic water through the Barents Sea but in reality, AW is not distinguished from other water masses, possibly present at 50 m. I strongly suggest including a definition of Atlantic water either in the introduction or in the methods.	m/s) for the AW inflow. For the calculation of the transports of NCC and central AW inflow, we use a spatial definition (NCC: 20°E, 70.1-71.45°N; central AW inflow: 20°E, 71.45-73.48°N).” (Il. 404-408) Regarding water mass definition we are of the opinion that in the Barents Sea, a T/S definition of AW would be ambiguous. As you note, the AW undergoes pronounced modification through heat loss during its passage through the BS. It is thus subject to constant modification which makes constraining the definition in T/S space challenging. However, as you suggest, this water is still of Atlantic origin, which might be a better term. We modified the caption of Fig. 4 to reflect this.
8	Line 51-52: '...the winter volume transport... ... estimated as 1.6+/-0.25 Sv, which appears high...' Observational estimates are in lines 44-45 are given for the annual means while an estimate based on a numerical simulation is calculated as a winter mean. While the argument about winter intensification is valid, a comparison between total annual means of AW transports from observations and model results would be more helpful.	This is a good point and we have added the annual means of the volume transports to the first results paragraph for a further multi-annual comparison between the model and observational estimates. “For the 1979-2019 period, the results generally show that the volume transport of AW inflow is stronger than that of the return-flow, with annual means of 0.9 Sv (NCC), 2.8 Sv (central AW inflow), and 1.2 Sv (return-flow) - a result consistent with observational estimates^{3,27}.” (Il. 111-114)

	Lines 55-56: '...which is yet unaccounted for in BSO AW (heat) transport estimates.' This part of the sentence is unclear. Do the authors mean that the return flow may significantly contribute to the variability of AW transport and thus to the variability of heat transport or something else?	Indeed, we mean that the return flow may significantly contribute to the variability of net AW transport between the Norwegian coast and Bear Island (full BSO section) and consequently may have an imprint in the variability of heat transport associated with the AW that was yet not considered as solely the inflow region was covered with long term moorings. We modified the sentence. It now reads: “These previous findings, however, suggest that the return-flow may significantly contribute to the interannual variability of the net AW transport through the BSO which is yet unaccounted for in observational BSO AW transport estimates which only consider the AW inflow regions^{9,24,13}.”
9	Lines 65-66: FESOM abbreviation should be explained when it appears for the first time (i.e., line 51).	Indeed, we modified the manuscript accordingly.
10	Line 74: '...AW temperature variability in the BSO inflow and return-flow...' Could the authors provide R (correlation coefficient) between observed and simulated time series also for AW temperature variability in the inflow and return flow (as it is done for sea ice area)? Since all three time series are	We have added the respective numbers to the manuscript. However, since the observational timeseries of temperature are not available each month as in the model but unevenly spaced in time these numbers should be interpreted with caution. In particular for winter means this might lead to just a single snapshot value from the observations being compared to a 4-month mean from the model. This has also been noted in the manuscript when supplying the correlation coefficients.

	included in Fig. 1, it would give better measure how well does the model reproduce AW temperature in the BSO.	The manuscript now reads: “The model demonstrates high skill in reproducing observed maximum annual sea ice area (SIA_{max}) variability in the BS (correlation: $R=0.92$, $p<0.01$) (Fig. 1c) and AW temperature variability (Fig. 1d, e) in the BSO AW inflow (monthly: $R=0.78$, winter: $R=0.59$, $p<0.01$) and return-flow (monthly: $R=0.57$, winter: $R=0.74$, $p<0.01$). It should, however, be noted that the temperature observations depict irregularly conducted snapshots in time whereas the model provides monthly means.” (ll. 98-104)
11	Lines 75-76: '...the volume transport variability of the AW-carrying currents...' This is unnecessary complication, I suggest 'the variability of AW volume transport...'	We have amended the lines accordingly.
12	Line 87: '...simulation based on the 1979-2019 period' It is a crucial information about the simulation period and it should be mentioned earlier when the numerical model is introduced. For the estimated AW transports in different branches based on observations, the averaging periods should be also given.	We have added the information on the full simulation period and the examined period to the beginning of the results section. It now reads: “For this study, we conducted a hindcast simulation with a configuration of FESOM2.1 that has been optimized for the Nordic Seas and Arctic Ocean. The simulation covers the period 1958-2019, of which the period 1979-2019 is examined in this study (see methods for model details).” (ll. 95-98)
13	Lines 89-90: '...the mean inflow and return-flow temperatures differ...'	These are the temperatures in the model at the locations where the observations are recorded. We have added this piece of information for clarity to the caption of Fig. SI1.

	Are these the mean temperatures of entire inflow/return flow or the mean temperatures of AW in the inflow/return flow?	
14	Line 97: 'With AW representing the dominant water mass...' Again, how is AW defined in the return flow? Does its definition account for cooling and freshening along the route in the Barents Sea?	We compute the AW inflow from the return-flow by separating the eastward from the westward flow, and do not separate AW from entrained water in the return-flow by making assumptions about the cooling and freshening of the inflow along its path. This has been clarified in the Methods section (and in the response to comment 7 above). The observational estimate of 80% is provided as a reference. Since the model return-flow temperature and observation-based return-flow temperature match well, we conclude that the 80% AW is a fairly good fit for the model too.
15	Line 107: '...of the same freezing season' It is unnecessary repetition since the season has been already defined (December-March).	We removed the respective statement.
	Lines 116-122: Correlations between SI _A max and AW volume transport in different branches. Since it has been well established that amount (variability) of ocean heat carried into the Barents Sea with Atlantic inflow is mostly driven by changes in AW temperature, not volume transport, it would be interesting to see whether also AW temperature in the return flow reflects	This is indeed an interesting point. The correlation between the detrended temperature anomalies of the return-flow and detrended anomalous SI _A max is negative, however, quite weak ($R=-0.42$, $p<0.01$). Hence, a warmer return-flow is correlated with reduced sea ice area. We added a respective statement: "The temperature of the return-flow is, in contrast, weakly anti-correlated with SI _A max ($R=-0.42$, $p<0.01$)." (ll. 172-173)

	SIAMax variability, i.e. if stronger cooling of the returning AW is correlated with an increased sea ice loss (due to more ocean heat lost to the ice).	
16	Lines 130-131: '...suggests an overall barotropic acceleration of the return-flow in winters with increased SIAMax' Since velocity is taken as predictor, the sentence would be more logical when formulated 'suggests increased SIAMax in winters with an overall barotropic acceleration of the return flow'.	We amended the lines accordingly.
17	Line 183: 'In years with increased SIAMax...' It rather should be 'decreased SIAMax' when stronger transport in the central Barents Sea provides more heat to the sea ice for melting or inhibiting ice formation.	We amended the lines accordingly.
18	Lines 195-197: 'This means the correlation is strongest downstream of the recirculation where the variability of the AW circulation should be strongest.' The meaning of this sentence is unclear, in particular its second part. Why the variability of AW circulations should be strongest (what does it mean - changing pathways, changing strength) downstream of its recirculation cell? If the AW	The sentence refers to when the recirculation is strong, less AW is transported further into the BS (and more AW transported into the BS when recirculation is weak). This is seen in the anomalies of the upper ocean velocity field of the BS that opposes the anomaly of the return-flow in Fig. 4. Increased re-circulation is associated with less AW throughflow through the BS. We made this clearer in the text. It now reads: "Thus, the correlation is strongest downstream of the recirculation where either more or less AW is present depending on the recirculation strength as

	recirculation changes, it implies changes in the downstream AW flow patterns but why should they be stronger than changes in recirculation? Also a statement in lines 200-203 about AW circulation variability and sea ice edge position is a bit obscure.	indicated by the anomalous upper ocean heat content (Fig. 4c, d).” (ll. 265-267)
19	Line 206: '...plays a major role...' In my opinion, a statement about 'a major role' is slightly overestimated since there is no objective comparison which driver of the sea ice area variability is a dominant one. I agree that the strength of AW recirculation is an important driver but not a 'major' one as it has not been compared to other forcings.	We agree with this point. Our intention here is not to precisely quantify the contribution of the return-flow to the sea ice variability but rather to point out that the return-flow is an important player which has simply been overlooked in this aspect so far, and that we found evidence that it should be taken into account. We have balanced all respective statements accordingly.
20	Line 215-216: '...there is a near-compensation...' What is a 'near-compensation'? Do the authors mean 'a near-surface compensation' or the fact that there is nearly full compensation between the return flow and the BSX outflow?	We clarified this in the text. It means that there is almost a full compensation of volume transport anomalies between the return-flow and the flow through BSX. In the BSO the anomalies are barotropic, in the BSX the anomalies reveal a baroclinic structure. (Fig. 3) The according paragraph now reads: “Indeed, the eastward volume transport through the BSX (from the BS into the Kara Sea) is strongly anti-correlated with SIA_{max} ($R=-0.69$, $p<0.01$). Repeating the regression analysis (Fig. 3b) with the velocity field along the BSX section reveals that the outflow out of the BS across the BSX section in winter (Fig. 3c) is weaker in winters with increased SIA_{max} (Fig. 3d). In contrast to the almost barotropic anomaly of the return-flow in the BSO (Fig. 3b), the regression

		analysis in the BSX reveals a surface-intensified response (Fig. 3d). This component of the flow yields the mass-compensating mechanism of anomalous return-flow events (Fig. S13). Our results thus demonstrate a barotropically strengthened westward return-flow in the BSO that is compensated by a baroclinic, upper ocean intensified, weakening of the eastward volume transport through the BSX. The quasi-simultaneous velocity/volume transport co-variability between the return-flow in the BSO and the currents in the BSX therefore suggests a connection along the AW pathways through the BS which is further explored in the following section.” (ll. 212-224)
21	Line 220-221: '...simulation over the 1979-2020 period...' In the preceding text, the simulation period is defined as 1979-2019.	We amended the respective statement.
22	Line 222: 'The complementary reduction...'. Why 'complementary' when it is entirely different process than the increase of AW temperature in the inflow?	We have removed the word “complementary”.
23	Line 254-255: '...how the flow anomaly is communicated upstream from the Svalbard shelf edge to the AW circulation branches in the BS...' This sentence is unclear. Do the author suggest that barotropic anomaly of the return flow velocity has its origin on the Svalbard shelf edge? What would be its linkage to the northern BSO?	The return-flow is located at the southern edge of Spitsbergen Bank, as illustrated in Fig. 1 and Fig. 3. An anomaly there, which assumingly triggers the observed patterns in the BS throughflow and in BSX, must be communicated throughout the BS. We assume that this is done by fast barotropic waves which travel along the Svalbard/northern BS shelf break towards BSX. Since this is more a suggestion than a sophisticated statement we leave the paragraph quite vague. We modified the paragraph a little. It now reads: “We consider it conceivable that changes in the return-flow driven by local wind anomalies, such as the intensity and frequency of synoptic-scale cyclones over Svalbard^{29,30,31}, may spin up or down the BSO return-flow depending on the

		wind direction. This is mainly supported by the barotropic nature of the velocity anomaly in the BSO return-flow in winters with anomalous SIA (Fig. 3b), typical of coast-parallel wind setting up an across-shelf sea surface height gradient which drives a geostrophic flow along the shelf edge. It is, however, unclear how the flow anomaly is communicated from the return-flow to the AW circulation branches within the BS that finally result in the coherent anomalous flow structure we revealed in the BS and up to BSX (Fig. 4a, b). Several studies have suggested the existence of a wind-excited eastward, topography-following passage of fast barotropic waves in the Eurasian part of the Arctic Ocean^{34,35}. It could thus be that an initial barotropic adjustment of the return-flow by wind-driven changes of the sea surface height excites barotropic waves, which travel eastward and adjust the AW throughflow. Addressing this question in more detail seems a promising subject of future study.” (ll. 319-336)
24	Lines 318-321: Annual Maximum Sea Ice Area. It should be mentioned that SI_{Amax} is taken for the month with maximum ice area which can be different for different years.	We added this to the paragraph. It now reads: “Annual Maximum Sea Ice Area Sea ice area of the BS is computed for every month in the 1979-2019 period. It is computed as the total area bounded by 18°E, 60°E, 68°N, 81°N in which the sea ice concentration exceeds 15%, for observational data (NSIDC, ERA5) and the model (FESOM2.1). Note that the month of annual maximum sea ice area is different for different years.” (ll. 410-414)
25	Lines 322-327: Linear regression Taking into account that on Fig. 4 regression coefficients are expressed in velocity units per standard deviation of SI_{Amax}, all time series are not only detrended and with their means	We added this to the paragraph. It now reads: “Linear Regression In this study, linear least squares regression is applied between the eastward velocity in the BSO/BSX and the SI_{Amax} of the BS. Before executing the respective regression fit, all time-series are linearly detrended in time-space and

	removed but also standardized in some way. This should be mentioned in the description.	the mean is removed. The timeseries of SIA_{max} is standardized by dividing the detrended anomalous SIA_{max} by its standard deviation. The statistical significance of the regression slopes is tested on a 99% confidence level based on a two-sided hypothesis test.” (ll. 415-421)
26	Line 333-335: Composite analysis The parameter which has been used in composite analysis shown on Fig. 4a,b should be indicated in the description. Of three parameters shown on the maps, two are actually winter means, not composites, and only one - anomaly of absolute velocity (is it absolute velocity or rather current speed?) - is a composite for extreme SIA_{max} years.	We have modified the entire Figure due to another reviewer's request. The problems you correctly address here are solved in the new Figure 4.
27	Figure 3: The meaning of dotted areas should be explained in the caption.	We have added a corresponding sentence to the caption.
28	Figure 4 a and 4 b: Colored isotherms overlaid on color shading for velocity anomaly are hardly legible in areas with stronger anomalies and make it difficult to distinguish the current vectors. In result, the plots are overcrowded even if the spatial distribution of temperature at 50 m is nowhere discussed in the paper. I would suggest removing mean temperature contours from the maps.	We have amended Fig. 4 (see comment above), and have removed the temperature contours and instead added bathymetry contours as the flow through the BS is bathymetrically guided.
29	Line 471: The reference to Ingvaldsen (2020) lacks a proper DOI. The full citation is: Randi	We have added the DOI to our bibliography.

	Ingvaldsen (2020) Mooring data from the Barents Sea Opening – Atlantic Water inflow https://doi.org/10.21335/NMDC-1838527821	
--	--	--

Comments on The manuscript title “Declining Atlantic Water Return-Flow Suppresses Wintertime Sea Ice Formation in the Barents Sea” by Finn Ole Heukamp and Claudia Wekerle et al.

Overall recommendation: Major revisions

This manuscript examines “**Declining Atlantic Water Return-Flow Suppresses Wintertime Sea Ice Formation in the Barents Sea**”. This topic is very interesting and meaningful. The author has provided a new perspective to understand what cause Arctic sea ice has declined in all seasons. I thought that this manuscript has potential to be published in Nature communications after provide more enough pieces of evidence in detail. Thus, I recommend that the manuscript needs major revision.

Major comments:

1. Author highlighted that the oceanic processes can impact on the Barents Sea (BS) sea ice decline by the different time scales. However, some studies also found that the atmospheric processes also influence the BS sea ice interannual variability, for example, the winter Barents-Kara Seas (BKS) sea ice associated with Ural blocking (UB) (Luo et al., 2017) and BKS sea ice variability show large interannual variations and related to the UB with the positive phase of North Atlantic Oscillation (NAO⁺) under the La Niña events (Luo et al., 2023) (-Although the author cites the article, there is no discussion about the article in the full text). Meanwhile, BS sea ice interannual and decadal variability also related to the North Atlantic atmospheric and oceanic transports (Shi et al., 2024). Arctic sea ice loss may also associated with the Arctic amplification under increasing CO₂ (Dai et al., 2019). So, I recommend that author need to add more discussion and references about the relationship between the atmospheric processes and arctic sea ice loss.
2. Why did author used the NSIDC instead of ERA5 dataset (eg: sea ice extent or sea ice cover)? And why did the author choose the time period of 1979-2019 instead of 1979-2022 or 1979-2023? Is it due to the limitation of the length of available sea ice area data or another reason? It is suggested that the author explain this reason why you choose the 1979-2019? (or supplement file with sea ice cover with ERA5 data dataset).
3. Line 65, “**Finite VolumE Sea Ice and Ocean Model (FESOM2.1)**” changed to “Finite Volume Sea Ice and Ocean Model (FESOM2.1)”.
4. Line 172, “STD” changed to “standard deviation (STD)?”.
5. Line 272, “79N” changed to “79°N”.
6. Line 300, Author used the JRA55-do atmosphere reanalysis for driving ocean sea ice model. Why author used the JRA55 instead of the ERA5 and NECP dataset? Is there any difference in this results between the two datasets, and it affect the overall results? Please give the ERA5 data or NCEP data results or add this results in supplementary file.
7. Line 334, Author used the SIA_{max} exceed ± 1 standard deviation. Why author do not used the ± 0.5 standard deviation SIA_{max}? Is there any difference in SIA_{max} and their composite results between ± 0.5 and ± 1 standard deviation? Please add the

composite results with the ± 0.5 standard deviation in supplementary file.

8. The authors used FESOM2.1 model to verify that return-flow determines AW volume and heat transport in the BSO region, then cause the SIC decline over the BSO region, Whether other models will have the same results? So I suggest that the authors need to consider other models to validate this results, such as using the MITgcm, CESM and Cmp6 models results, or add more discussion about the models results in discussion part.

Reference:

- Luo, B., D. Luo, L. Wu, L. Zhong, and I. Simmonds, 2017: Atmospheric circulation patterns which promote winter Arctic sea ice decline. *Environ. Res. Lett.*, 12, 054017, <https://doi.org/10.1088/1748-9326/aa69d0>
- Luo, B., Luo, D., Ge, Y. et al. Origins of Barents-Kara sea-ice interannual variability modulated by the Atlantic pathway of El Niño–Southern Oscillation. *Nat Commun* 14, 585 (2023). <https://doi.org/10.1038/s41467-023-36136-5>
- Shi, J., Luo, B., Luo, D. et al. Differing roles of North Atlantic oceanic and atmospheric transports in the winter Eurasian Arctic sea-ice interannual-to-decadal variability. *npj Clim Atmos Sci* 7, 62 (2024). <https://doi.org/10.1038/s41612-024-00605-5>
- Dai, A., Luo, D., Song, M. et al. Arctic amplification is caused by sea-ice loss under increasing CO₂. *Nat Commun* 10, 121 (2019). <https://doi.org/10.1038/s41467-018-07954-9>